# Latent Bayesian Optimization via Autoregressive Normalizing Flows

**Seunghun Lee**[1], **Jinyoung Park**[1], **Jaewon Chu**[1], **Minseo Yoon**[1], **Hyunwoo J. Kim**[2]*

[1]Department of Computer Science and Engineering, Korea University
[2]School of Computing, KAIST
`{llsshh319,lpmn678,allonsy07,cooki0615}`@korea.ac.kr
`hyunwoojkim`@kaist.ac.kr

## Abstract

Bayesian Optimization (BO) has been recognized for its effectiveness in optimizing expensive and complex objective functions. Recent advancements in Latent Bayesian Optimization (LBO) have shown promise by integrating generative models such as variational autoencoders (VAEs) to manage the complexity of high-dimensional and structured data spaces. However, existing LBO approaches often suffer from the *value discrepancy problem*, which arises from the reconstruction gap between input and latent spaces. This value discrepancy problem propagates errors throughout the optimization process, leading to suboptimal outcomes. To address this issue, we propose a Normalizing Flow-based Bayesian Optimization (NF-BO), which utilizes normalizing flow as a generative model to establish *one-to-one* encoding function from the input space to the latent space, along with its left-inverse decoding function, eliminating the reconstruction gap. Specifically, we introduce SeqFlow, an autoregressive normalizing flow for sequence data. In addition, we develop a new candidate sampling strategy that dynamically adjusts the exploration probability for each token based on its importance. Through extensive experiments, our NF-BO method demonstrates superior performance in molecule generation tasks, significantly outperforming both traditional and recent LBO approaches.

## 1 Introduction

Bayesian optimization (BO) (Kushner, 1962; 1964) has been broadly applied across various areas such as chemical design (Wang & Dowling, 2022), material science (Ament et al., 2021), and hyperparameter optimization (Wu et al., 2019). BO aims to probabilistically optimize an expensive and black-box objective function using a surrogate model to find an optimal solution with minimal cost. Although BO is effective in continuous spaces, its application to a discrete input space still remains challenging (Oh et al., 2019; Deshwal & Doppa, 2021). Latent Bayesian Optimization (LBO) (Gómez-Bombarelli et al., 2018; Tripp et al., 2020) addresses this challenge by performing BO in a lower-dimensional latent space learned by a generative model such as Variational AutoEncoders (VAEs) (Kingma & Welling, 2014). LBO performs optimization in a continuous space by mapping the discrete input into a continuous latent space with the VAEs (Kusner et al., 2017; Jin et al., 2018; Samanta et al., 2019).

However, the reconstruction of VAE is not always perfect, leading to *value discrepancy problem*, which indicates that given a sample encoded as an embedding in the latent space, its decoding may not result in the same sample in the input space. Figure 1 shows the value discrepancy problem by presenting the distributions of objective values before and after the reconstruction using a pretrained SELFIES VAE (Maus et al., 2022), focusing on data with the top 10% of objective values.

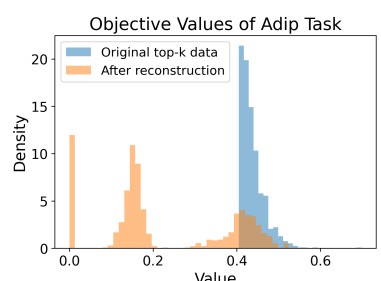

Figure 1: Visualization of value discrepancy problem.

---

*Corresponding author

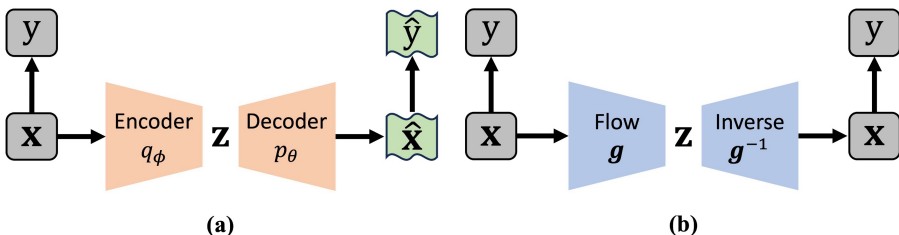

Figure 2: **(a)** Most existing LBO approaches suffer from the value discrepancy problem $y \neq \hat{y}$ induced by the reconstruction gap, $p_\theta(q_\phi(\mathbf{x})) \neq \mathbf{x}$. This results in that the latent representation $\mathbf{z}$ corresponds to different evaluation values $y$ and $\hat{y}$ due to the reconstruction error, where $\mathbf{x} \neq \hat{\mathbf{x}}$. **(b)** Our NF-BO effectively addresses the value discrepancy problem by employing a normalizing flow model that ensures one-to-one mapping between $\mathbf{x}$ and $\mathbf{z}$ via the invertible flow and inverse processes, $\boldsymbol{g}$ and $\boldsymbol{g}^{-1}$, *i.e.*, $\boldsymbol{g}^{-1}(\boldsymbol{g}(\mathbf{x})) = \mathbf{x}$. So, the latent representation $\mathbf{z}$ is consistently associated with the same evaluation value $y$.

During the optimization process, these models often refine the latent space by training on newly searched data and their corresponding objective values (Maus et al., 2022; Lee et al., 2023). It requires re-encoding data points to find their latent representations, which also makes the value discrepancy problem. The previous method (Chu et al., 2024) addressed this by inverse the data with an iterative approach.

To address the problems efficiently without re-evaluations/iterative procedures, we propose a Normalizing Flows-based Bayesian Optimization, referred to as NF-BO, that leverages an invertible function for discrete sequence data. This approach establishes a one-to-one encoding function from the input space to the latent space, along with its left-inverse decoding function, effectively resolving the value discrepancy problem.

Figure 2 explains the value discrepancy problem in (a) and how our NF-BO model addresses it using flow and inversion (b). Apart from the value discrepancy problem, we additionally introduce token-level adaptive candidate sampling for more effective local search. The sampling scheme dynamically adjusts the sampling distribution based on the importance of each token to more focus on promising areas.

The contributions of our research are as follows.

- We propose NF-BO to address the value discrepancy problem, which commonly occurs in Latent Bayesian Optimization (LBO). NF-BO leverages normalizing flows to establish a one-to-one encoding function from the input space to the latent space, with its left-inverse decoding function ensuring accurate reconstruction. To the best of our knowledge, NF-BO is the first work to integrate normalizing flows into LBO.

- We propose a Token-level Adaptive Candidate Sampling (TACS), enabling effective local search by adjusting the sampling distribution based on the token-level importance.

- Our extensive experiments on multiple benchmarks demonstrate the superiority of the proposed method in optimizing high-dimensional and structured data, consistently outperforming existing latent Bayesian optimization and traditional optimization methods.

## 2 RELATED WORKS

**Latent Bayesian Optimization.** Latent Bayesian Optimization (LBO) (Gómez-Bombarelli et al., 2018; Eissman et al., 2018; Tripp et al., 2020; Griffiths & Hernández-Lobato, 2020; Grosnit et al., 2021; Siivola et al., 2021) has emerged as an effective approach to overcome the limitations of traditional Bayesian Optimization (BO), particularly in high-dimensional or discrete input spaces. By embedding discrete sequences into a continuous latent space, typically using Variational Autoencoders (VAEs) (Kingma & Welling, 2014; Higgins et al., 2017), LBO enables efficient optimization of complex problems, as discussed in (González-Duque et al., 2024) with a comprehensive review.

To improve this mapping, prior works have proposed novel architectures to improve reconstruction quality (Kusner et al., 2017; Jin et al., 2018; Lu et al., 2018; Samanta et al., 2019) or utilize uncertainty for increased robustness (Notin et al., 2021; Verma et al., 2022). In particular, LaMBO (Stanton et al., 2022) introduced a masked language model-based architecture, and LaMBO-2 (Gruver et al., 2024) developed a diffusion-based approach to extend prior methods.

Recent LBO works, such as LOL-BO (Maus et al., 2022) have introduced the concept of trust regions (Eriksson et al., 2019) in the latent space. ROBOT (Maus et al., 2023) have emphasized the importance of incorporating diversity measures to further support diverse solutions. CoBO (Lee et al., 2023) implements a novel loss function to improve the alignment between the latent space and the objective function. However, these methods still encounter the value discrepancy problem, where the output value from the decoded input is inconsistent with the original value.

**Normalizing Flows.** Normalizing Flows (NFs) (Rezende & Mohamed, 2015) are a class of generative models that transform a simple, known probability distribution into a more complex one and *vice versa*. Each layer in these models is designed to be invertible, with a tractable Jacobian determinant, which facilitates efficient computation and flexible modeling of complex data distributions. Early NF models (Dinh et al., 2015; 2017; Kingma & Dhariwal, 2018; Ho et al., 2019; Durkan et al., 2019) have demonstrated their effectiveness in generating high-quality images using coupling-based techniques, ensuring tractability and scalability.

More recently, NFs have also been developed not only for generating images but also for expanding their applicability to a wider range of data types. For instance, methods like (Ziegler & Rush, 2019) specifically addressed the challenges in modeling discrete data by integrating NFs within a VAE framework, jointly learning latent distributions and improving the expressivity of the latent space. To the best of our knowledge, our work is the first work that applies NFs in the context of LBO to deal with the value discrepancy problem by introducing a new model SeqFlow in Section 4.2.

## 3 PRELIMINARIES

**Bayesian optimization (BO)** has widely been applied to optimize black-box (unknown) objective functions where their evaluations are expensive. Let $\mathcal{X}$ and $\mathbf{x}$ be the input space and a solution, respectively. The goal of BO is to find the optimal solution $\mathbf{x}^*$ that maximizes a black-box objective function $f$, which can be formulated as:

$$\mathbf{x}^* = \arg\max_{\mathbf{x} \in \mathcal{X}} f(\mathbf{x}). \tag{1}$$

Since $f$ is unknown, BO typically constructs a surrogate model $\hat{f}$ to approximate the true function $f$. With the surrogate model, BO searches for the optimal points with an acquisition function $\alpha$ as follows:

$$\tilde{\mathbf{x}} = \arg\max_{\mathbf{x} \in \mathcal{X}_{\text{cand}}} \alpha(\mathbf{x}; \hat{f}, \mathcal{D}), \tag{2}$$

where $\mathcal{D} = \{(\mathbf{x}^{(i)}, y^{(i)})\}_{i=1}^{N}$ represents the accumulated data, $\tilde{\mathbf{x}}$ is a data point selected based on the acquisition function, and $\mathcal{X}_{\text{cand}} \subseteq \mathcal{X}$ is a candidate set. In trust region-based local Bayesian optimization such as TuRBO (Eriksson et al., 2019), $\mathcal{X}_{\text{cand}}$ is selected within a trust region that is often centered at a current optimal point (*e.g.*, anchor point). The trust region limits the search space to promising small regions, thereby easing the difficulty of optimization.

**Normalizing Flows (NFs)** (Rezende & Mohamed, 2015) are a class of generative models for modeling the data distributions $p(\mathbf{x})$ through a sequence of invertible transformations, offering exact density evaluation and sample generation. NFs are formulated as follows:

$$\mathbf{z} = g(\mathbf{x}; \theta), \quad \mathbf{x} = g^{-1}(\mathbf{z}; \theta), \tag{3}$$

where $g$ and $g^{-1}$ denote the forward and inverse transformation, parameterized by $\theta$, ensuring that each mapping is bijective and differentiable. The determinant of Jacobian $|\det J_g(\mathbf{x})|^{-1}$ computes the change in volume induced by $g$, which is important for density calculations. The training of these flows involves minimizing the following negative log-likelihood:

$$\mathcal{L} = -\mathbb{E}_{\mathbf{x} \sim \mathcal{X}} [\log p(\mathbf{x})] = -\mathbb{E}_{\mathbf{x} \sim \mathcal{X}} \left[ \log p(\mathbf{z}) + \log \left| \det \frac{\partial g}{\partial \mathbf{x}} \right| \right]. \tag{4}$$

This ensures that the model accurately captures the underlying data distribution, allowing efficient generation.

# 4 METHODS

We propose Normalizing Flow-based Bayesian Optimization (NF-BO), which leverages Normalizing Flows (NFs) as a generative model combined with adaptive candidate sampling for effective optimization. To begin with, we introduce Latent Bayesian Optimization (LBO) and the value discrepancy problem induced by incomplete reconstruction of the generative model used in LBO (Section 4.1). Next, we present an autoregressive NF model, SeqFlow, specifically tailored for sequence generation, which addresses the value discrepancy problem by accurate reconstruction (Section 4.2). Additionally, we propose Token-level Adaptive Candidate Sampling (TACS), which constructs a diverse candidate set within trust regions (Section 4.3). Finally, we delineate the overall process of our NF-BO (Section 4.4).

## 4.1 PROBLEM STATEMENT

Although BO has shown its effectiveness in various optimization tasks, it has difficulty performing over the discrete domain, such as chemical design (Griffiths & Hernández-Lobato, 2020; Wang & Dowling, 2022). To address this issue, recent works (Gómez-Bombarelli et al., 2018; Tripp et al., 2020) have studied Latent Bayesian Optimization (LBO) that performs BO in a continuous latent space after embedding the discrete input data into the latent space. LBO can be formulated as:

$$\mathbf{z}^* = \operatorname*{argmax}_{\mathbf{z} \in \mathcal{Z}} f(p_\theta(\mathbf{z})), \tag{5}$$

where $\mathcal{Z}$ is a latent space and $p_\theta : \mathcal{Z} \mapsto \mathcal{X}$ is the decoder parameterized by $\theta$. LBO uses an encoder-decoder structure to map complex inputs into an effective representation in the latent space and then performs a search in this latent space. Note that the formulation assumes the decoder $p_\theta$ is deterministic. LBO searches for the optimal points using acquisition function $\alpha$ as follows:

$$\tilde{\mathbf{x}} = p_\theta(\tilde{\mathbf{z}}), \text{ where } \tilde{\mathbf{z}} = \operatorname*{argmax}_{\mathbf{z} \in \mathcal{Z}_{\mathrm{cand}}} \alpha(\mathbf{z}; \hat{f}, \mathcal{D}). \tag{6}$$

$\mathcal{D} = \{(\mathbf{x}^{(i)}, \mathbf{z}^{(i)}, y^{(i)})\}_{i=1}^N$ represents the accumulated data, $\tilde{\mathbf{x}}$ and $\tilde{\mathbf{z}}$ are the next evaluation point and its corresponding latent vector in the candidate set $\mathcal{Z}_{\mathrm{cand}} \subseteq \mathcal{Z}$. $\hat{f} : \mathcal{Z} \mapsto \mathcal{Y}$ is a surrogate model for the composite function $f \circ p_\theta : \mathcal{Z} \mapsto \mathcal{Y}$.

**Value Discrepancy Problem.** LBOs generally learn a surrogate model in the latent space and construct the data $\{(\mathbf{x}^{(i)}, \mathbf{z}^{(i)}, y^{(i)})\}_{i=1}^N$, where $y^{(i)} = f(\mathbf{x}^{(i)})$, $\mathbf{z}^{(i)} = q_\phi(\mathbf{x}^{(i)})$, with the encoder $q_\phi$, assuming complete reconstruction $\mathbf{x}^{(i)} = p_\theta(q_\phi(\mathbf{x}^{(i)}))$ and identical function values, *i.e.*, $y^{(i)} = f(\mathbf{x}^{(i)}) = f(p_\theta(\mathbf{z}^{(i)}))$ (Tripp et al., 2020; Maus et al., 2022; Lee et al., 2023; Chen et al., 2024). However, in practice, there exists a reconstruction gap in VAE and it results in the discrepancy between the function values evaluated at input data $\mathbf{x}$ and its reconstruction $\hat{\mathbf{x}}$ as follows:

$$\mathbf{x} \neq \hat{\mathbf{x}} \text{ and } f(\mathbf{x}) \neq f(\hat{\mathbf{x}}), \text{ where } \hat{\mathbf{x}} := p_\theta(q_\phi(\mathbf{x})). \tag{7}$$

This value discrepancy problem propagates errors throughout the optimization process, leading to suboptimal optimization results. To mitigate this issue, an ideal generative model in LBO should exhibit perfect reconstruction, ensuring that any point in the input space can be accurately mapped to the latent space and *vice versa*. This property resolves the value discrepancy problem by ensuring that the generated data accurately reflects the characteristics of the original data. As a result, error propagation during optimization is minimized, leading to improved optimization performance. Motivated by this, we introduce a new LBO built on normalizing flows.

## 4.2 SEQFLOW

To address the value discrepancy problem in existing LBOs, we propose Normalizing Flow-based Bayesian Optimization (NF-BO), leveraging NF's ability in modeling the data distribution via a one-to-one mapping between the input space and the latent space. To efficiently perform NF-BO on a long sequence of discrete data, we propose a novel discrete **Seq**uence-specialized autoregressive normalizing **Flow** model (**SeqFlow**).

SeqFlow learns the distribution $p(\mathbf{x})$ of the sequence of discrete data $\mathbf{x} = [\mathbf{x}_1, \ldots, \mathbf{x}_L]$, where $\mathbf{x} \in \mathbb{N}^L$ is a sequence of token indices, using two components: **(i)** a mapping function between

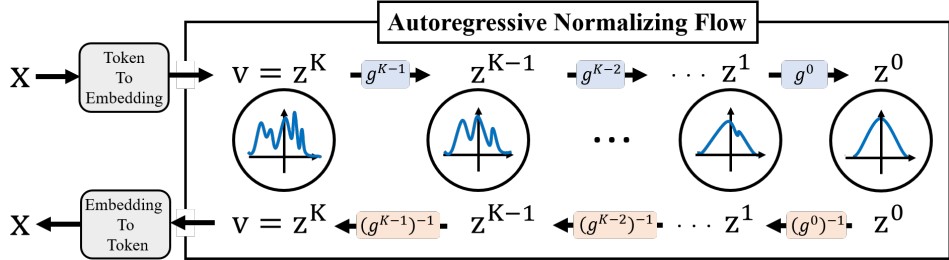

Figure 3: **Overall pipeline of SeqFlow**. Given the input space of a sequence discrete values $\mathbf{x}$, SeqFlow first maps the discrete values $\mathbf{x}$ to continuous representation $\mathbf{v}$ and efficiently transforms them via autoregressive transformations $\{g^i\}_{i=0}^{K-1}$ to a latent representation $\mathbf{z}^0$ in the encoding phase (top pathway). In the decoding phase (bottom pathway), SeqFlow reconstructs $\mathbf{x}$ from $\mathbf{z}^0$ through the inverse of transformations. SeqFlow ensures the perfect reconstruction of the discrete input.

the continuous representation $\mathbf{v} \in \mathbb{R}^{L \times F}$ and a discrete input $\mathbf{x}$ and **(ii)** a density model $p(\mathbf{v})$ (*i.e.*, normalizing flow). Here, $L$ represents the number of tokens in a sequence and $F$ is the embedding dimension. The mapping function **(i)** is defined as:

$$\mathbf{x}_i = \arg\max_j \ \mathrm{sim}\left(\mathbf{v}_i, \mathbf{e}_j\right), \tag{8}$$

where $\mathrm{sim}(\cdot, \cdot)$ is the cosine similarity, $\mathbf{e}_j \in \mathbb{R}^F$ is an embedding vector of $j$-th token. All embeddings are initialized by random vectors drawn from a normal distribution after L2 normaliztion, *i.e.*, $\|\mathbf{e}_j\|_2 = 1$ for all $j$. As a result, $\mathbf{x}_i$ is the index of the token whose embedding vector $\mathbf{e}_j$ is most similar to the continuous representation vector $\mathbf{v}_i$. Based on the density model $p(\mathbf{v})$ and the mapping function, we define the likelihood of input discrete sequence $p(\mathbf{x})$ as follows:

$$p(\mathbf{x}) = \int p(\mathbf{v}) \prod_i^L p(\mathbf{x}_i|\mathbf{v}_i) \, d\mathbf{v},$$
$$p(\mathbf{x}_i|\mathbf{v}_i) = \delta_{\mathbf{x}_i, \check{\mathbf{x}}_i}, \text{where } \check{\mathbf{x}}_i = \arg\max_j \ \mathrm{sim}(\mathbf{v}_i, \mathbf{e}_j), \tag{9}$$

where $\delta$ is the Kronecker delta function and $\check{\mathbf{x}}_i$ is the index of the most similar embedding vector to $\mathbf{v}_i$. However, directly calculating Eq. (9) is intractable. So, we introduce the variational distribution $q(\mathbf{v}_i|\mathbf{x}_i)$ (Ho et al., 2019) and optimize the likelihood $p(\mathbf{x})$ by maximizing the evidence lower bound (ELBO), which is derived as:

$$\log p(\mathbf{x}) \geqslant \mathbb{E}_{\mathbf{v}_1 \sim q(\mathbf{v}_1|\mathbf{x}_1), \dots, \mathbf{v}_L \sim q(\mathbf{v}_L|\mathbf{x}_L)} \left[ \log p(\mathbf{v}) + \sum_i^L \left(\log p(\mathbf{x}_i|\mathbf{v}_i) - \log q(\mathbf{v}_i|\mathbf{x}_i)\right) \right]. \tag{10}$$

We define the distribution $q(\mathbf{v}_i|\mathbf{x}_i)$ as an isotropic Gaussian distribution centered at the embedding of $\mathbf{x}_i$, *i.e.*, $\mathcal{N}(\mathbf{e}_{\mathbf{x}_i}, \sigma^2 I)$. Additionally, we sample only $\mathbf{v}_i$ from $q(\mathbf{v}_i|\mathbf{x}_i)$ that satisfies $p(\mathbf{x}_i|\mathbf{v}_i) = 1$. The constrained version of $q(\mathbf{v}_i|\mathbf{x}_i)$ is defined as:

$$q'(\mathbf{v}_i|\mathbf{x}_i) = \begin{cases} \frac{q(\mathbf{v}_i|\mathbf{x}_i)}{Z}, & \text{if } p(\mathbf{x}_i|\mathbf{v}_i) = 1 \\ 0, & \text{otherwise} \end{cases}, \tag{11}$$

where $Z$ is a normalization constant. We accept a sample $\mathbf{v}_i$ with probability $\frac{q'(\mathbf{v}_i|\mathbf{x}_i)}{q(\mathbf{v}_i|\mathbf{x}_i)/Z}$. Through the constrained sampling within the domain where the condition holds, we effectively make the practical sampling distribution $q(\mathbf{v}_i|\mathbf{x}_i)$ closer to $p(\mathbf{v}_i|\mathbf{x}_i)$. The example of the distribution $q'$ is depicted in the Appendix G.

We employ a negative log likelihood to maximize $\log p(\mathbf{v})$, which serves as a normalizing flow loss that enhances the model's ability to generate valid continuous representations $\mathbf{v}$. The Negative Log-Likelihood $\mathcal{L}_{\mathrm{NLL}}$ is defined as follows:

$$\mathcal{L}_{\mathrm{NLL}} = -\log p(\mathbf{v}) = -\log p(\mathbf{z}) - \sum_{k=0}^{K-1} \log \left| \det \frac{\partial g^k}{\partial \mathbf{z}^{k+1}} \right|, \tag{12}$$

where $g^k$ represents $k$-th transformation in the flow sequence $\boldsymbol{g}$ and $\mathbf{z}^{k+1}$ is the output of the $k$-th transformation.

Also, we implement a simple variant of the contrastive loss to maximize the cosine similarity between $\mathbf{v}_i$ and $\mathbf{e}_{\mathbf{x}_i}$ for $\mathbf{x}_i$ and distance it from other embeddings:

$$\mathcal{L}_{\text{sim}}(\mathbf{v}, \mathbf{e}) = -\frac{1}{L}\sum_{i=1}^{L} \text{sim}(\mathbf{v}_i, \mathbf{e}_{\mathbf{x}_i}) + \frac{1}{L}\sum_{i=1}^{L} \text{sim}(\mathbf{v}_i, \mathbf{e}_j), \ \mathbf{e}_j \sim \text{Unif}(\mathcal{E}\backslash\{\mathbf{e}_{\mathbf{x}_i}\}), \qquad (13)$$

where $\mathbf{e}_j$ is an embedding uniformly sampled from embedding set $\mathcal{E}$ except for $\mathbf{e}_{\mathbf{x}_i}$, which corresponds to the token $\mathbf{x}_i$. The contrastive loss encourages diverse token embeddings in a given context. To train our SeqFlow model, we combine the similarity loss with the Negative Log-Likelihood (NLL) loss of normalizing flows. The final loss of our model is given by:

$$\mathcal{L}_{\text{NF-BO}} = \mathcal{L}_{\text{NLL}} + \lambda\mathcal{L}_{\text{sim}}(\mathbf{v}, \mathbf{e}), \qquad (14)$$

where $\lambda$ is the hyperparameter that balances the NLL loss and the similarity loss.

**Autoregressive Normalizing Flows.** To effectively represent a long sequence of discrete values, we adopt an autoregressive normalizing flows (Ziegler & Rush, 2019). Our model defines the flow for encoding:

$$\mathbf{v} = \boldsymbol{g}^{-1}(\mathbf{z}; \theta), \quad \mathbf{z} = \boldsymbol{g}(\mathbf{v}; \theta), \qquad (15)$$

where $\boldsymbol{g}$, $\boldsymbol{g}^{-1}$ are entire flow and its inverse transformation, respectively. To be specific, autoregressive NF is composed of $K$ series of autoregressive transformation blocks and each block for $k \in \{0, \ldots, K-1\}$ operates as follows:

$$\mathbf{z}_i^{k+1} = (g^k)^{-1}\left(\mathbf{z}_i^k; \mathbf{z}_{<i}^{k+1}, \theta^k\right), \text{ and } \mathbf{z}_i^k = g^k\left(\mathbf{z}_i^{k+1}; \mathbf{z}_{<i}^{k+1}, \theta^k\right), \qquad (16)$$

where $\mathbf{z}_i^k$ denotes $i$-th token output vector of the $k$-th block. The initial input to the first block is $\mathbf{z}^0 = \mathbf{z}$, and the output of the final block is $\mathbf{z}^K = \mathbf{v}$. Our autoregressive block $(g^k)^{-1}$ consists of several coupling layers, which aggregate information from the previous tokens. This helps the flow model to capture the long-range dependencies within the sequence for effective sequence modeling. More details on the architecture of the autoregressive normalizing flow model is in the Appendix H.

**Injectivity of our SeqFlow.** The SeqFlow ensures injectivity through the invertibility of the transformation function $\boldsymbol{g}$. This function maps the embedding $\mathbf{e}_{\mathbf{x}}$ to a latent representation $\mathbf{z}$, and the decoding process serves as the left inverse of this encoding. As stated in Proposition 1 and Proposition 2, this guarantees that for every input $\mathbf{x}$, the operation $\boldsymbol{g}(\mathbf{e}_{\mathbf{x}})$ and its inverse will precisely reconstruct $\mathbf{x}$.

**Proposition 1.** *Let $g$ be Normalizing Flows and $h$ is an injective function with a nonempty domain $\mathcal{X}$. Then, $f := g \circ h$ is left invertible, i.e., $f^{-1} \circ f = id_X$, where $h^{-1}$ is the left inverse of $h$ and $f^{-1} := h^{-1} \circ g^{-1}$.*

*Remarks.* Proposition 1 implies that our construction provides perfect reconstruction. To be specific, SeqFlow consists of two functions: (i) a function $h$ to map a discrete sequence data to a sequence of embeddings in the continuous space and (ii) Normalizing Flows $g$ defined in the continuous space. If the function $h$ is injective, with its left inverse and the inverse of NFs, SeqFlow achieves the *perfect reconstruction*.

**Proposition 2.** *Assume the elements of embedding set $\mathcal{E} = \{\mathbf{e}_1, \mathbf{e}_2, \ldots, \mathbf{e}_{|\mathcal{E}|}\}$ are distinct and L2-normalized, i.e., $\mathbf{e}_i \neq \mathbf{e}_j$, for all $i \neq j$ and $\|\mathbf{e}_i\|_2 = 1$. Given a list of $L$ natural numbers $\mathbf{x} = [\mathbf{x}_1, \mathbf{x}_2, \ldots, \mathbf{x}_L] \in \mathbb{N}^L$, a mapping function $h$ is defined as $h(\mathbf{x}) := \mathbf{e}_{\mathbf{x}}$ where $\mathbf{e}_{\mathbf{x}} = [\mathbf{e}_{\mathbf{x}_1}, \mathbf{e}_{\mathbf{x}_2}, \ldots, \mathbf{e}_{\mathbf{x}_L}]^T$. Then, $h$ is injective and the function $h^{-1}(\mathbf{v}) := [\arg\max_j \text{sim}(\mathbf{v}_i, \mathbf{e}_j)]_{i=1}^{L}$, where $sim(\mathbf{e}_i, \mathbf{e}_j) = \mathbf{e}_i^T\mathbf{e}_j$, is a left inverse of $h$, i.e., $h^{-1}(h(\mathbf{x})) = \mathbf{x}$.*

The proofs of Proposition 1 and Proposition 2 are provided in Appendix E. This approach ensures all information is preserved during encoding and decoding through a one-to-one function and its left-inverse. This is crucial for applications that demand exact input reconstruction.

Moreover, the reliability of the decoding function $h(\mathbf{z})$ ensures that any generated latent variable accurately reverts to its corresponding input sequence. This capacity is essential for resolving the value

discrepancy problem often observed in other latent-based optimization models, where reconstructed outputs might not match the original inputs. This enhancement increases the overall efficacy of the optimization process, making SeqFlow a robust framework for handling discrete sequence optimization tasks.

### 4.3 Token-level Adaptive Candidate Sampling

In this section, we present a Token-level Adaptive Candidate Sampling (TACS) to improve the candidate sampling process of trust region-based local search BO methods (Eriksson et al., 2019; Maus et al., 2022; Lee et al., 2023). These local search BO methods search next query points constrained in promising areas centered around an anchor points, derived from the best input found in the data history.

Most previous trust-region-based approaches utilize Thompson sampling on a finite set of candidate points $\mathbf{Z}_{\text{cand}}$ by perturbing a subset of dimensions of an anchor point (Eriksson et al., 2019). We observe that the existing approaches select a subset of dimensions to be perturbed *uniformly*, which can lead to less effective exploration especially when it is applied to our SeqFlow. To address this, we propose Token-level Adaptive Candidate Sampling (TACS), which samples candidates regarding the importance of each latent token. Specifically, we sample a subset of latent tokens of an anchor point for perturbation from a token-level probability distribution, defined by the relative importance of each token. This allows TACS to perform a dense search over important tokens while sparsely exploring less important ones with limited resources.

To identify important tokens at the anchor point $(\mathbf{x}, \mathbf{z})$, we utilize the Pointwise Mutual Information (PMI) between each token $\mathbf{z}_i$ and the sequence $\mathbf{x}$.

$$\omega_i(\mathbf{z}) = \text{PMI}(\mathbf{x}, \mathbf{z}_i | \mathbf{z}_{-i}) = \log \frac{p(\mathbf{x}|\mathbf{z})}{p(\mathbf{x}|\mathbf{z}_{-i})} = \log \frac{p(\mathbf{x}|\mathbf{z})}{\mathbb{E}_{\mathbf{z}_i \sim \mathcal{N}(\mathbf{0}, I)}(p(\mathbf{x}|\mathbf{z}))},$$

$$p(\mathbf{x}|\mathbf{z}) = p(\mathbf{x}|\mathbf{v}) = \prod_i^L p(\mathbf{x}_i|\mathbf{v}_i), \tag{17}$$

where $\mathbf{z}_{-i} = \{\mathbf{z}_1, \mathbf{z}_2, \ldots, \mathbf{z}_{i-1}, \mathbf{z}_{i+1}, \ldots, \mathbf{z}_L\}$. A Monte Carlo approximation is employed to estimate $p(\mathbf{x}|\mathbf{z}_{-i})$, and to stabilize computations, a small constant $\epsilon$ is added to $p(\mathbf{x}_i|\mathbf{v}_i)$. This PMI score $\omega_i(\mathbf{z})$ measures the impact of latent token $\mathbf{z}_i$ on the sequence $\mathbf{x}$, enabling efficient exploration along the most important dimensions. Using the PMI score, we define the token-level sampling probability $\pi_i(\mathbf{z})$ as:

$$\pi_i(\mathbf{z}) = \min\left(\kappa s_i(\mathbf{z}), 1\right), \quad s_i(\mathbf{z}) = \frac{\exp\left(\omega_i(\mathbf{z})/\tau\right)}{\sum_j \exp\left(\omega_j(\mathbf{z})/\tau\right)}, \tag{18}$$

where $\kappa$ is a constant scaling factor, and $\tau$ indicates the temperature. The softmax with temperature $\tau$ allows for flexible adjustment in focusing on the importance of different tokens. For example, if $\tau$ has a higher value, the candidate set is uniformly sampled, disregarding the token-level importance. Conversely, a lower $\tau$ concentrates sampling more densely on the tokens with the highest importance.

### 4.4 Overall Bayesian Optimization Process

In this section, we present our overall NF-BO framework, which is illustrated in Figure 4. For each iteration, the NF-BO framework begins by training the SeqFlow model with the loss function $\mathcal{L}_{\text{NF-BO}}$ as defined in Eq. (14), using the dataset $\mathcal{D} = \{(\mathbf{x}^{(i)}, y^{(i)})\}$. For training the SeqFlow model, we sample variational vector $\mathbf{v}$ following the distribution $q'$, as described in Eq. (11). After training the SeqFlow, we construct the latent vector $\mathbf{z}^{(i)}$ corresponding to the input $\mathbf{x}^{(i)}$ and then use it to train the surrogate model $\hat{f}$. Then, we select anchor points $\mathbf{z}_{\text{anc}}$ based on their corresponding objective values $y$ and generate trust regions centered on them. To perform local search, the candidate set $\mathcal{Z}_{\text{cand}}$ is drawn within the trust region, using the Token-level Adaptive Candidate Sampling (TACS) method. Finally, the acquisition function $\alpha$ determines next query point $\tilde{\mathbf{z}}$ followed by decoding and evaluating it to update the best score. This procedure is repeated until the allocated oracle budget $T$ is expended, continuously improving the SeqFlow model throughout the optimization process. For better understanding, the pseudocode for NF-BO is provided in the Appendix F.

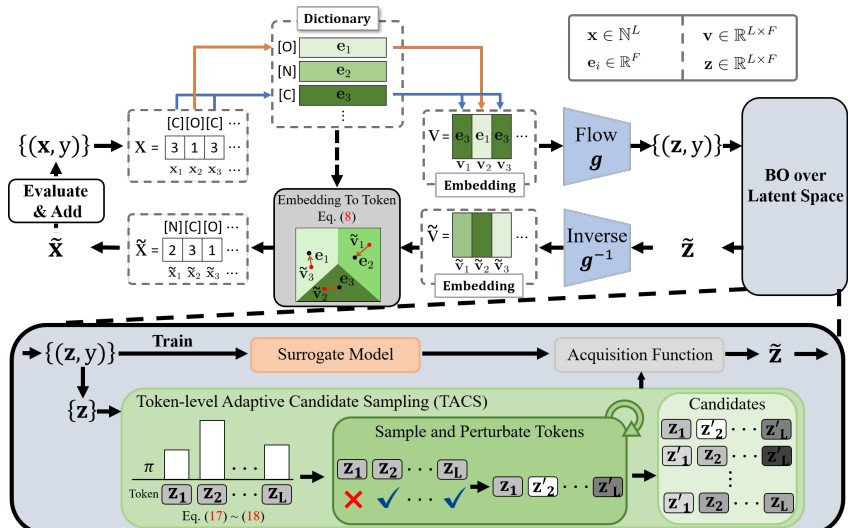

Figure 4: **Overview of NF-BO**. We employ our normalizing flows, **SeqFlow**, as a mapping function between a discrete input space and a continuous latent space. Each discrete input token $\mathbf{x}_i$ is mapped to its corresponding embedding vector $\mathbf{v}_i$ from the dictionary. A surrogate model is then trained using the latent representation $\mathbf{z}$ encoded by the flow model $g$ and the associated function value $y$ to emulate the objective function. To enhance the efficiency of trust region-based local search, we propose a **Token-level Adaptive Candidate Sampling (TACS)**. In TACS, candidates for the acquisition function are generated by perturbing tokens, sampled according to a token-level sampling probability $\pi$, specified in Eq. (18). Given these candidates and the surrogate model, we select the next query points $\tilde{\mathbf{z}}$ by the acquisition function. Next, the inverse model $g^{-1}$ generates the embedding $\tilde{\mathbf{v}}$ and searches the most similar embedding and return the corresponding index as a $\tilde{\mathbf{x}}$.

## 5 EXPERIMENTS

### 5.1 TASKS

We validate our NF-BO across various benchmarks focusing on *de novo* molecular design tasks. Initially, we conduct experiments on the Guacamol benchmarks (Brown et al., 2019), specifically targeting seven challenging tasks where optimal solutions are not readily found. For these benchmarks, we evaluate NF-BO and the baselines under three different settings, each varying the number of initial data points and the additional oracle budget: (100, 500), (10,000, 10,000), and (10,000, 70,000). Subsequently, we evaluate our method on the PMO benchmarks (Gao et al., 2022), which consists of 23 tasks, including albuterol similarity, and amlodipine MPO.

### 5.2 BASELINES

In the Guacamol benchmark, we use LSBO, TuRBO-$L$ (Eriksson et al., 2019), W-LBO (Tripp et al., 2020), LOLBO (Maus et al., 2022), CoBO (Lee et al., 2023), and PG-LBO (Chen et al., 2024) as the baselines. In the PMO benchmarks, we compare our method with 25 molecular design algorithms. These include generative models (*e.g.*, GANs and VAEs), machine learning models (*e.g.*, Reinforcement Learning), and optimization algorithms (*e.g.*, MCTS and GA). More detailed explanations of the baselines are in Appendix J.

### 5.3 IMPLEMENTATION DETAILS

We employ Thompson sampling (Eriksson et al., 2019) as the acquisition function, and our surrogate model is a sparse variational Gaussian process (Snelson & Ghahramani, 2005; Hensman et al., 2015; Matthews, 2017) enhanced with a deep kernel (Wilson et al., 2016). For the Guacamol and PMO benchmarks, we pretrain using 1.27M unlabeled Guacamol and 250K ZINC datasets, respectively,

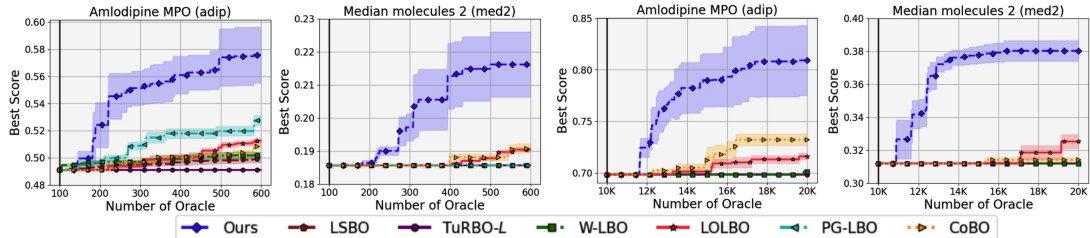

Figure 5: Optimization results of NF-BO on Guacamol benchmarks comparing performance with baselines under two oracle budget settings: (100, 500) (left) and (10,000, 10,000) (right). The shaded regions indicate the standard error over 5 trials.

Table 1: PMO results across various methods and assembly. The table presents scores and rankings for 6 evaluation metrics illustrating the comparative performance of each method. Score is the sum of all 23 tasks constituting the PMO benchmark computed to summarize the overall performance.

| Methods | Assembly | Top-1 Score (Rank) | Top-10 Score (Rank) | Top-100 Score (Rank) | AUC Top-1 Score (Rank) | AUC Top-10 Score (Rank) | AUC Top-100 Score (Rank) |
|---|---|---|---|---|---|---|---|
| *Bayesian Optimization* | | | | | | | |
| **NF-BO** | **SELFIES** | **18.095 (1)** | **17.692 (1)** | **17.037 (1)** | **15.539 (1)** | **14.737 (1)** | 13.423 (2) |
| GP BO | Fragments | 15.345 (7) | 14.940 (6) | 14.365 (6) | 13.798 (5) | 13.156 (5) | 12.122 (6) |
| VAE BO | SELFIES | 11.423 (17) | 9.788 (19) | 7.622 (22) | 10.589 (17) | 8.887 (19) | 6.899 (22) |
| VAE BO | SMILES | 10.926 (21) | 9.435 (21) | 7.623 (21) | 10.197 (19) | 8.587 (21) | 6.909 (21) |
| JT-VAE BO | Fragments | 10.296 (23) | 8.671 (24) | 7.037 (24) | 9.973 (22) | 8.358 (24) | 6.740 (23) |
| *Reinforcement Learning* | | | | | | | |
| REINVENT | SMILES | 16.772 (2) | 16.654 (2) | 16.297 (2) | 14.711 (2) | 14.196 (2) | **13.445 (1)** |
| REINVENT | SELFIES | 16.059 (5) | 15.889 (4) | 15.377 (3) | 14.077 (4) | 13.471 (4) | 12.475 (5) |
| MolDQN | Atoms | 7.143 (26) | 6.495 (26) | 5.435 (26) | 6.332 (26) | 5.620 (26) | 4.528 (26) |
| *Genetic Algorithm* | | | | | | | |
| Graph GA | Fragments | 16.244 (4) | 15.946 (3) | 15.342 (4) | 14.356 (3) | 13.751 (3) | 12.696 (3) |
| STONED | SELFIES | 14.257 (8) | 14.201 (8) | 14.017 (7) | 13.256 (7) | 13.024 (6) | 12.518 (4) |
| SMILES GA | SMILES | 13.123 (11) | 12.997 (9) | 12.824 (9) | 12.357 (10) | 12.054 (8) | 11.598 (7) |
| SynNet | Synthesis | 13.105 (12) | 12.279 (12) | 10.768 (15) | 12.425 (9) | 11.498 (9) | 9.914 (9) |
| GA+D | SELFIES | 11.942 (16) | 11.696 (15) | 11.230 (13) | 9.387 (24) | 8.964 (18) | 8.280 (15) |
| *Hill Climbing* | | | | | | | |
| LSTM HC | SMILES | 16.754 (3) | 15.880 (5) | 14.621 (5) | 13.611 (8) | 12.223 (7) | 10.365 (8) |
| LSTM HC | SELFIES | 13.770 (9) | 12.894 (10) | 11.657 (12) | 11.441 (14) | 10.246 (15) | 8.595 (13) |
| DoG-Gen | Synthesis | 15.633 (6) | 14.772 (7) | 13.653 (8) | 12.721 (8) | 11.456 (10) | 9.635 (12) |
| MIMOSA | Fragments | 12.524 (15) | 12.223 (13) | 11.717 (11) | 11.378 (15) | 10.651 (13) | 9.708 (11) |

following the previous settings. We employ 1,000 initial data points and an additional 9,000 oracle calls following the PMO benchmarks.

## 5.4 RESULTS ON GUACAMOL BENCHMARKS

We compare the optimization results of our NF-BO with six LBO baselines in two experimental settings: 500 and 10K additional oracle budgets on two Guacamol tasks. Figure 5 presents the main experimental results, while the other results on five tasks are provided in Appendix B. The experimental results demonstrate that our proposed NF-BO consistently outperforms other VAE-based LBO methods in all tasks and settings.

## 5.5 RESULTS ON PMO BENCHMARKS

We also conduct experiments to demonstrate the effectiveness of our NF-BO against 25 baseline models, including various generative models and optimization algorithms, across 23 PMO benchmark tasks. Our evaluation metrics included Top-1, Top-10, and Top-100 scores, as well as the Area

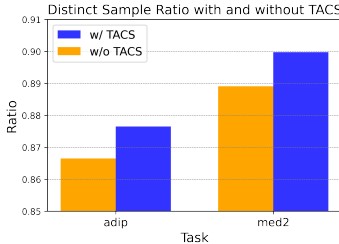 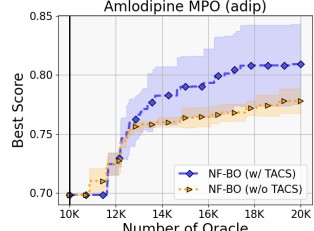 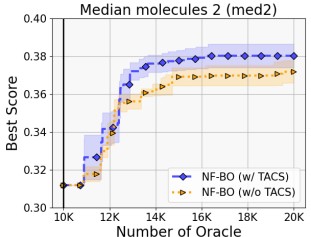

Figure 6: Comparison of distinct sample ratios with and without TACS in two Guacamol tasks.

Figure 7: Comparison of performance with and without TACS in Guacamol benchmarks. The shaded regions indicate the standard error over 5 trials.

Under the Curve (AUC) for these metrics, all based on Oracle calls. The experimental results are in Table 1.

The scores for each individual task are detailed in Appendix A. The table shows that our NF-BO achieves the best performance with 1st rank on five out of six metrics. In particular, NF-BO significantly enhances the performance of VAE BO, which also uses SELFIES, improving its average rank from 19th to 1st.

# 6    ANALYSIS

In this section, we provide the analysis of our NF-BO on the Guacamol. Experiments were implemented with 10,000 initial data points and an additional oracle budget of 10,000.

## 6.1    CANDIDATE DIVERSITY WITH TACS IMPLEMENTATION

We evaluate the proportion of distinct samples within a set of 1,000 candidates generated in two different Guacamol tasks with and without TACS. Each experimental setup was subjected to Monte Carlo approximation 10 times to estimate expectation in Eq. (17), and we conducted five independent experiments averaging the results. We use a pre-trained SeqFlow model and 10 different anchor points to generate trust regions.

In Figure 6, the result with TACS has a higher ratio of distinct samples compared to those without TACS, underscoring its effectiveness in enhancing the diversity of the candidate pool. This implies TACS improves the exploration capacity of the BO, which is crucial for optimization performance. We provide optimization performances with different temperatures in TACS in Appendix C.

## 6.2    ABLATION STUDY

Figure 7 our ablation studies that illustrates the effectiveness of our Token-level Adaptive Candidate Sampling (TACS) strategy, shows its impact on performance across these tasks in the Guacamol benchmark. From the analysis, it is evident that the incorporation of TACS significantly enhances performance, confirming its benefit in optimizing the search process.

# 7    CONCLUSION

In conclusion, the proposed NF-BO method, which leverages normalizing flows, makes significant improvements in the domain of Bayesian optimization, especially for handling molecular data. This approach not only addresses the value discrepancy problem through a one-to-one function from the input space to the latent space and its left-inverse function but also enhances the effectiveness of the search process with a novel token-level adaptive candidate sampling strategy. Our comprehensive evaluations across diverse benchmarks have demonstrated the superiority of NF-BO over traditional methods and other LBO techniques, confirming its potential to reshape the landscape of optimization strategies in various scientific and engineering applications.

REPRODUCIBILITY STATEMENT

For reproducibility, we elaborate on the overall pipeline of our work in Section 4. In our main paper and appendix, we also illustrate our overall pipeline and pseudocode for NF-BO, respectively. Code is available at https://github.com/mlvlab/NFBO.

ETHICS STATEMENT

Our main contribution, NF-BO, aims to design molecules with desired properties, *e.g.,* Amlodipine MPO in the Guacamol task. However, this could lead to unintended consequences, such as the creation of harmful substances like illicit drugs, requiring the exercise of extreme caution.

ACKNOWLEDGEMENT

This research was supported by the ASTRA Project through the National Research Foundation (NRF) funded by the Ministry of Science and ICT (No. RS-2024-00439619).

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

Table 2: Detailed results on PMO benchmarks. The table presents scores and standard deviations across 6 evaluation metrics, with each score representing the mean of 5 independent runs. Additionally, the sum for each column is computed to summarize the overall performance.

| | Top-1 | Top-10 | Top-100 | AUC Top-1 | AUC Top-10 | AUC Top-100 |
|---|---|---|---|---|---|---|
| albuterol_similarity | $1.000 \pm 0.000$ | $0.967 \pm 0.011$ | $0.847 \pm 0.035$ | $0.862 \pm 0.014$ | $0.817 \pm 0.010$ | $0.708 \pm 0.021$ |
| amlodipine_mpo | $0.802 \pm 0.028$ | $0.798 \pm 0.024$ | $0.788 \pm 0.013$ | $0.688 \pm 0.023$ | $0.672 \pm 0.021$ | $0.642 \pm 0.020$ |
| celecoxib_rediscovery | $0.799 \pm 0.164$ | $0.699 \pm 0.076$ | $0.634 \pm 0.063$ | $0.605 \pm 0.069$ | $0.546 \pm 0.031$ | $0.481 \pm 0.024$ |
| deco_hop | $0.725 \pm 0.006$ | $0.724 \pm 0.007$ | $0.724 \pm 0.007$ | $0.685 \pm 0.004$ | $0.675 \pm 0.003$ | $0.662 \pm 0.003$ |
| drd2 | $1.000 \pm 0.000$ | $1.000 \pm 0.000$ | $0.999 \pm 0.001$ | $0.932 \pm 0.004$ | $0.875 \pm 0.005$ | $0.788 \pm 0.004$ |
| fexofenadine_mpo | $0.854 \pm 0.012$ | $0.854 \pm 0.012$ | $0.852 \pm 0.012$ | $0.797 \pm 0.008$ | $0.784 \pm 0.008$ | $0.756 \pm 0.007$ |
| gsk3b | $0.990 \pm 0.015$ | $0.952 \pm 0.041$ | $0.903 \pm 0.069$ | $0.820 \pm 0.032$ | $0.754 \pm 0.010$ | $0.664 \pm 0.028$ |
| isomers_c7h8n2o2 | $1.000 \pm 0.000$ | $0.841 \pm 0.076$ | $0.619 \pm 0.160$ | $0.916 \pm 0.005$ | $0.748 \pm 0.062$ | $0.525 \pm 0.126$ |
| isomers_c9h10n2o2pf2cl | $0.946 \pm 0.028$ | $0.935 \pm 0.008$ | $0.933 \pm 0.007$ | $0.881 \pm 0.010$ | $0.842 \pm 0.009$ | $0.757 \pm 0.009$ |
| jnk3 | $0.894 \pm 0.052$ | $0.884 \pm 0.061$ | $0.866 \pm 0.076$ | $0.709 \pm 0.036$ | $0.649 \pm 0.037$ | $0.574 \pm 0.040$ |
| median1 | $0.422 \pm 0.022$ | $0.419 \pm 0.023$ | $0.409 \pm 0.021$ | $0.352 \pm 0.007$ | $0.340 \pm 0.006$ | $0.307 \pm 0.004$ |
| median2 | $0.313 \pm 0.022$ | $0.311 \pm 0.021$ | $0.305 \pm 0.019$ | $0.269 \pm 0.013$ | $0.260 \pm 0.011$ | $0.244 \pm 0.010$ |
| mestranol_similarity | $0.758 \pm 0.058$ | $0.758 \pm 0.058$ | $0.758 \pm 0.058$ | $0.629 \pm 0.028$ | $0.607 \pm 0.024$ | $0.570 \pm 0.018$ |
| osimertinib_mpo | $0.880 \pm 0.010$ | $0.878 \pm 0.010$ | $0.872 \pm 0.012$ | $0.838 \pm 0.004$ | $0.828 \pm 0.005$ | $0.788 \pm 0.005$ |
| perindopril_mpo | $0.678 \pm 0.034$ | $0.678 \pm 0.034$ | $0.677 \pm 0.034$ | $0.598 \pm 0.028$ | $0.586 \pm 0.027$ | $0.560 \pm 0.026$ |
| qed | $0.948 \pm 0.000$ | $0.948 \pm 0.000$ | $0.948 \pm 0.000$ | $0.943 \pm 0.000$ | $0.941 \pm 0.000$ | $0.931 \pm 0.000$ |
| ranolazine_mpo | $0.844 \pm 0.012$ | $0.843 \pm 0.011$ | $0.838 \pm 0.009$ | $0.723 \pm 0.012$ | $0.698 \pm 0.010$ | $0.647 \pm 0.008$ |
| scaffold_hop | $0.769 \pm 0.172$ | $0.767 \pm 0.170$ | $0.733 \pm 0.141$ | $0.646 \pm 0.087$ | $0.629 \pm 0.087$ | $0.608 \pm 0.085$ |
| sitagliptin_mpo | $0.764 \pm 0.075$ | $0.757 \pm 0.079$ | $0.722 \pm 0.090$ | $0.578 \pm 0.032$ | $0.516 \pm 0.029$ | $0.427 \pm 0.025$ |
| thiothixene_rediscovery | $0.639 \pm 0.121$ | $0.623 \pm 0.100$ | $0.602 \pm 0.084$ | $0.524 \pm 0.061$ | $0.496 \pm 0.048$ | $0.459 \pm 0.037$ |
| troglitazone_rediscovery | $0.476 \pm 0.040$ | $0.475 \pm 0.039$ | $0.473 \pm 0.039$ | $0.386 \pm 0.020$ | $0.375 \pm 0.019$ | $0.352 \pm 0.018$ |
| valsartan_smarts | $0.998 \pm 0.001$ | $0.996 \pm 0.002$ | $0.974 \pm 0.012$ | $0.633 \pm 0.041$ | $0.594 \pm 0.037$ | $0.514 \pm 0.033$ |
| zaleplon_mpo | $0.593 \pm 0.016$ | $0.584 \pm 0.016$ | $0.561 \pm 0.016$ | $0.524 \pm 0.011$ | $0.504 \pm 0.011$ | $0.460 \pm 0.010$ |
| **Sum** | **18.095** | **17.692** | **17.037** | **15.539** | **14.737** | **13.423** |

## A  DETAILED RESULTS ON PMO BENCHMARKS

We conducted experiments to demonstrate the effectiveness of our NF-BO across 23 PMO benchmark tasks. The full experimental results, including detailed scores and standard deviations for each task, are provided in Table 2. The evaluation metrics we used include Top-1, Top-10, and Top-100 scores, as well as the Area Under the Curve (AUC) for these metrics, all based on oracle calls. Our main findings show that NF-BO consistently achieves competitive performance across various tasks. Additionally, the AUC scores show comparable results in terms of further highlighting NF-BO's robustness. These results suggest that NF-BO not only excels at identifying the best solutions but also maintains consistent performance across different tasks.

## B  ADDITIONAL RESULTS ON GUACAMOL BENCHMARKS

As referenced in Section 5.4, we compare our NF-BO with six LBO baselines across seven tasks in the Guacamol benchmarks. In this section, we present the results of the remaining tasks for the (100, 500) and (10,000, 10,000) oracle settings, which were not covered in the main section, along with the results for the (10,000, 70,000) oracle settings. Figures 8, 9, and 10 display the results for the (100, 500), (10,000, 10,000), and (10,000, 70,000) oracle settings, respectively. In the case of PG-LBO (Chen et al., 2024), we were unable to include results for the (10,000, 10,000) and (10,000, 70,000) settings due to infeasibility caused by excessive experimental time.

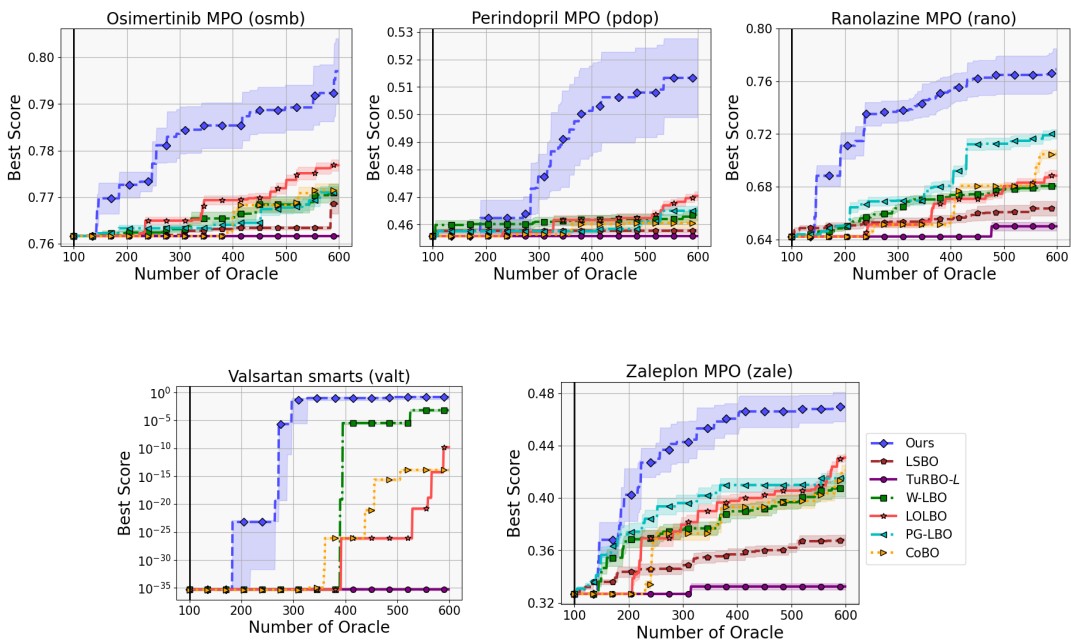

Figure 8: Optimization results on Guacamol benchmarks under 500 additional oracle settings. Note that in the valt task, the y-axis is represented on a log scale, and for this task, we also added two nonzero data points in the initial dataset of 100 for all methods. The shaded regions indicate the standard error over 5 trials.

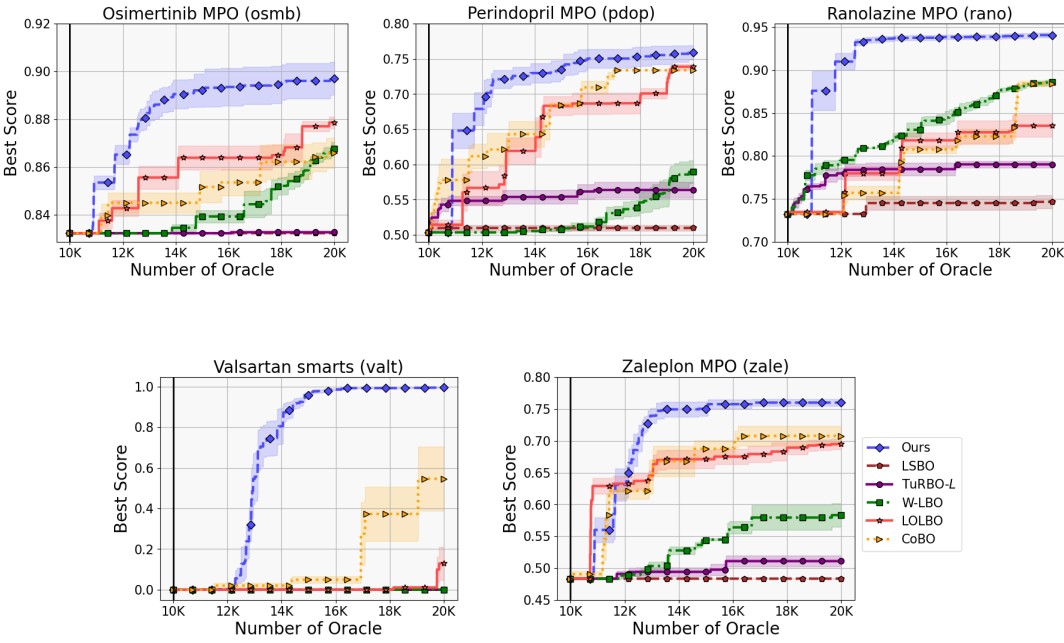

Figure 9: Optimization results on Guacamol benchmarks under 10K additional oracle settings. The shaded regions indicate the standard error over 5 trials.

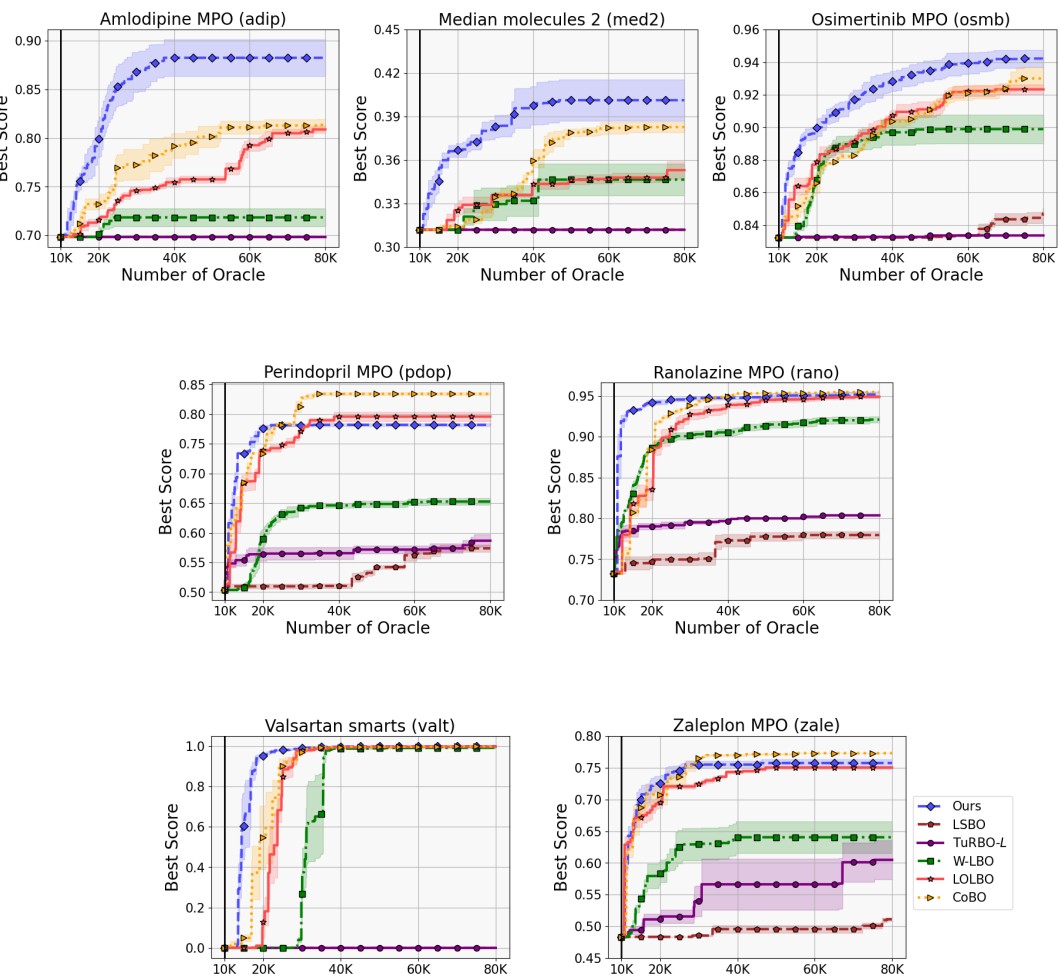

Figure 10: Optimization results on Guacamol benchmarks under 70K additional oracle settings. The shaded regions indicate the standard error over 5 trials.

## C    Distinct Sample Ratio with Various TACS Temperature

The distinct sample ratio quantifies the diversity of generated candidates by measuring the proportion of distinct samples within the total candidates. In Figure 11, we explore how varying the temperature parameter in TACS affects this ratio on the Guacamol benchmark. Lower temperatures generally promote exploration by sampling impactful tokens within the input sequences in the latent space, increasing the diversity of candidates.

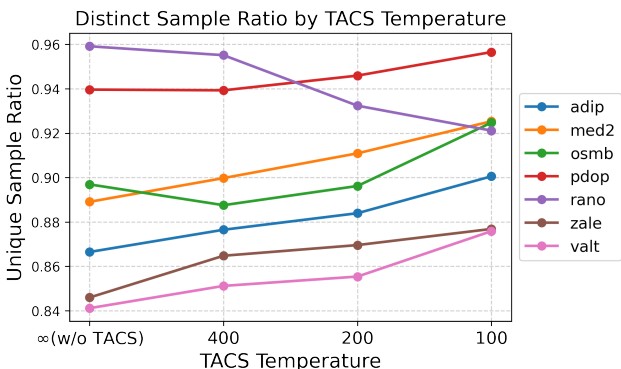

Figure 11: Distinct sample ratio with various TACS temperatures on Guacamol benchmarks.

The experimental setup follows the same configuration as detailed in the analysis section of the paper. As a result, we observe that for six of the seven tasks (excluding Rano), the distinct sample ratio increases as the temperature decreases, indicating that lower temperatures encourage a broader exploration of distinct candidates.

## D    Analysis of Pointwise Mutual Information in SeqFlow

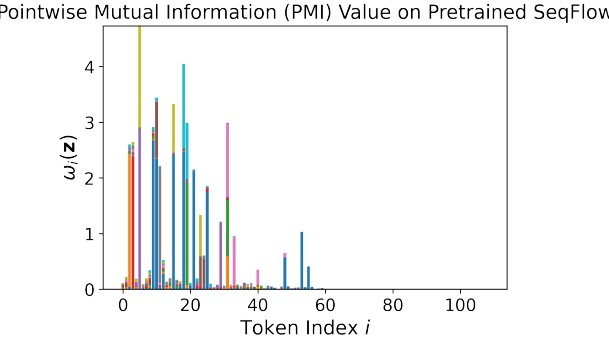

Figure 12: Pointwise Mutual Information (PMI) value $\omega_i$ between each latent token $\mathbf{z}_i$ and sequence $\mathbf{x}$.

We analyzed the Pointwise Mutual Information (PMI) values of each latent token $\mathbf{z}_i$ across different points in the sequence. The PMI values, denoted as $\omega_i(\mathbf{z}) = \mathrm{PMI}(\mathbf{x}, \mathbf{z}_i | \mathbf{z}_{-i})$, were calculated at 10 different points, and Monte Carlo methods were employed 10 times to ensure accuracy. The x-axis in Figure 12 represents the token index $i$ in the latent $\mathbf{z}$, while the y-axis measures the PMI value between each $\mathbf{z}_i$ and $\mathbf{x}$. Different colors stacked in the figure represent the cumulative PMI values measured from various points.

As observed in Figure 12, there is a trend where the PMI values decrease as the token index increases. This tendency reflects the autoregressive nature of our model used, where earlier tokens tend to influence a larger part of the sequence, exerting significant impacts on subsequent tokens. This shows that early tokens in our model are important to the sequence generation and optimization processes.

## E    PROOF OF LEFT INVERTIBILITY OF SEQFLOW

**Proposition 1.** *Let $g$ be Normalizing Flows and $h$ is an injective function with a nonempty domain $\mathcal{X}$. Then, $f := g \circ h$ is left invertible, i.e., $f^{-1} \circ f = id_X$, where $h^{-1}$ is the left inverse of $h$ and $f^{-1} := h^{-1} \circ g^{-1}$.*

*Proof.* NF $g$ has an inverse function $g^{-1}$ by definition and $h$ has a left inverse since every injective function with a nonempty domain has a left inverse. Let $h^{-1}$ denote the left inverse of $h$. Then, $f^{-1} \circ f := h^{-1} \circ g^{-1} \circ g \circ h = \mathrm{id}_X$. $\qquad\square$

**Proposition 2.** *Assume the elements of embedding set $\mathcal{E} = \{\mathbf{e}_1, \mathbf{e}_2, \ldots, \mathbf{e}_{|\mathcal{E}|}\}$ are distinct and L2-normalized, i.e., $\mathbf{e}_i \neq \mathbf{e}_j$, for all $i \neq j$ and $\|\mathbf{e}_i\|_2 = 1$. Given a list of $L$ natural numbers $\mathbf{x} = [\mathbf{x}_1, \mathbf{x}_2, \ldots, \mathbf{x}_L] \in \mathbb{N}^L$, a mapping function $h$ is defined as $h(\mathbf{x}) := \mathbf{e_x}$ where $\mathbf{e_x} = [\mathbf{e}_{\mathbf{x}_1}, \mathbf{e}_{\mathbf{x}_2}, \ldots, \mathbf{e}_{\mathbf{x}_L}]^T$. Then, $h$ is injective and the function $h^{-1}(\mathbf{v}) := [\arg\max_j \mathrm{sim}(\mathbf{v}_i, \mathbf{e}_j)]_{i=1}^L$, where $\mathrm{sim}(\mathbf{e}_i, \mathbf{e}_j) = \mathbf{e}_i^T \mathbf{e}_j$, is a left inverse of $h$, i.e., $h^{-1}(h(\mathbf{x})) = \mathbf{x}$.*

*Proof.* Since a function with a nonempty domain is injective if and only if the function has a left inverse, we show that $h^{-1}$ is the left inverse of $h$.

By definition, we have

$$h^{-1}(h(\mathbf{x})) = h^{-1}(\mathbf{e_x}) = \left[\arg\max_j \mathrm{sim}(\mathbf{e}_{\mathbf{x}_i}, \mathbf{e}_j)\right]_{i=1}^L. \tag{19}$$

Since the embeddings are distinct and L2-normalized, $\mathrm{sim}(\mathbf{e}_i, \mathbf{e}_j) = \mathbf{e}_i^T \mathbf{e}_j$ satisfies

$$\mathbf{e}_i^T \mathbf{e}_j = \begin{cases} 1, & \text{if } i = j, \\ < 1, & \text{otherwise.} \end{cases} \tag{20}$$

Thus, for each $i$, the maximum value of $\mathrm{sim}(\mathbf{e}_{\mathbf{x}_i}, \mathbf{e}_j)$ occurs at $j = \mathbf{x}_i$, meaning $\arg\max_j \mathrm{sim}(\mathbf{e}_{\mathbf{x}_i}, \mathbf{e}_j) = \mathbf{x}_i, \forall i$. Therefore, $h^{-1}(h(\mathbf{x})) = h^{-1}(\mathbf{e_x}) = [\mathbf{x}_i]_{i=1}^L = \mathbf{x}$. $\qquad\square$

## F    PSEUDOCODE OF NF-BO

This section provides the pseudocode of NF-BO frameworks on Algorithm 1. $topk$ in the algorithm refers to selecting the top $k$ data points with the highest objective values from the dataset $\mathcal{D}$. The number of data $k$ is specified in Table 3.

## G    VISUALIZATION OF SAMPLING DISTRIBUTION: FEASIBLE REGIONS IN LATENT SPACE

Figure 13 illustrates the simplified example of constrained sampling distribution $q'(\mathbf{v}_i|\mathbf{x}_i)$ based on Eq. (11). In the figure, the Voronoi cells represent the spatial partitioning of the input space. For a simple and clear description, this space is based on random points. Each cell is shaded based on an isotropic Gaussian distribution centered at the cell's origin. The shading intensity reflects the density of accepting a sample based on the condition $p(\mathbf{x}_i|\mathbf{v}_i) = 1$. Darker regions indicate higher Gaussian values, and hence higher likelihoods of sample acceptance. Light sky blue areas indicate regions with lower density compared to the darker regions. This visualization demonstrates the selective nature of our sampling method, focusing only on feasible solutions during optimization.

## H    ARCHITECTURE DETAILS

Each autoregressive block $g^k$ includes several coupling layers $g^{k,l}$. The transformation of each layer operates as follows:

$$\mathbf{z}_i^{k,l} = g^{k,l}\left(\mathbf{z}_i^{k,l+1}; A(\mathbf{z}_{<i}^{k,L}), \theta^{k,l}\right). \tag{21}$$

---

**Algorithm 1** NF-BO

---

**Input**: black-box objective function $f$, SeqFlow model $\boldsymbol{g}$, embedding set $\mathcal{E} = \{\mathbf{e}_1, \mathbf{e}_2, \ldots, \mathbf{e}_{|\mathcal{E}|}\}$, surrogate model $\hat{f}$, acquisition function $\alpha$, token-level importance $\omega$, oracle budget $T$, number of query points $N_q$, initial data $\mathcal{D} = \{(\mathbf{x}^{(i)}, y^{(i)})\}_{i=1}^{n}$

1: **for** $t = 1, 2, \ldots$, while the oracle budget remains **do**
2:      $\mathcal{D}_{tr} \leftarrow \text{CONCAT}\left(\mathcal{D}[-N_q :], topk(\mathcal{D})\right)$
3:      Train $\boldsymbol{g}, \mathcal{E}$ with $\mathcal{L}_{\text{NF-BO}}, \mathcal{D}_{tr}$                       $\triangleright$ *Eq. (14)*
4:      Train $\hat{f}$ on $\mathcal{D}_{tr}$ if $t \neq 1$ else $\mathcal{D}$
5:      $(\mathbf{x}_{\text{anc}}, y_{\text{anc}}) \leftarrow$ sample based on $y$ values from $\mathcal{D}$
6:      $\mathbf{z}_{\text{anc}} \leftarrow \boldsymbol{g}(\mathbf{e}_{\mathbf{x}_{\text{anc}}})$
7:      $\mathbf{Z}_{\text{cand}} \leftarrow$ Draw $N_q$ candidate points with TACS in trust region centered on $\mathbf{z}_{\text{anc}}$    $\triangleright$ *Eq. (18)*
8:      Select subset $\tilde{\mathbf{Z}}$ based on $\alpha(\mathbf{z}; \hat{f})$, where $\mathbf{z} \in \mathbf{Z}_{\text{cand}}$
9:      $\tilde{\mathbf{X}} \leftarrow \left\{\mathbf{x} | \mathbf{x} = [\arg\max_j \ \text{sim}(\mathbf{v}_i, \mathbf{e}_j)]_{i=1}^{L}, \mathbf{v} = \boldsymbol{g}^{-1}(\mathbf{z}), \mathbf{z} \in \tilde{\mathbf{Z}}\right\}$
10:     $\mathcal{D}_{\text{new}} \leftarrow \left\{(\mathbf{x}, f(\mathbf{x})) | \mathbf{x} \in \tilde{\mathbf{X}}\right\}$
11:     $\mathcal{D} \leftarrow \text{CONCAT}(\mathcal{D}, \mathcal{D}_{\text{new}})$
12: **end for**
13: $(\mathbf{x}^*, \mathbf{z}^*, y^*) \leftarrow \arg\max_{(\mathbf{x}, \mathbf{z}, y) \in \mathcal{D}} y$
14: **return** $\mathbf{x}^*$

---

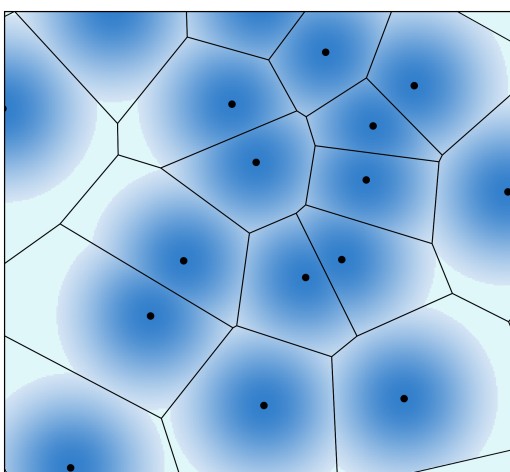

Figure 13: **Sampling Distribution Visualization.** Voronoi cells represent different regions, and color intensity indicates the likelihood of accepting a sample.

For each block $g^k$, the input is represented by $\mathbf{z}^{k,0} = \mathbf{z}^k$, and the output of the final layer in each block sets the initial condition for the next block, $\mathbf{z}^{k,L} = \mathbf{z}^{k+1,0}$. The final output after the last layer of the last block is $\mathbf{z}^{K,L} = \mathbf{v}$. Each coupling layer further refines the data representation, informed by previous tokens. The function $A$, which we implemented as an LSTM, aggregates information from prior tokens, enhancing the model's ability to capture long-range dependencies of sequence data.

## I   Implementation Details

In our experiments, parameters were adjusted based on the specific requirements of each benchmark setting. For (the batch size of trust regions, the number of query points $N_q$ per trust region), we set these parameters to (5, 10) for the Guacamol benchmark with an additional oracle call setting of 500. For other settings, these parameters were adjusted to (10, 100).

We explored the temperature $\tau$ for the Token-level Adaptive Candidate Sampling (TACS) across the values $\{400, 200, 100\}$ to find the optimal setting. The sequence length $L$ was determined based on the longest sequence in the initial dataset. For details on the other fixed parameters, please refer to Table 3.

Table 3: Fixed parameters for all tasks and settings.

| Parameter | Value |
|---|---|
| Scaling factor $\kappa$ in TACS | $0.1 \cdot$ Sequence length $L$ |
| Standard deviation $\sigma$ of variational distribution $q$ | 0.1 |
| # of topk data for training | 1000 |
| Coefficient of similarity loss $\mathcal{L}_{\mathrm{sim}}$ | 1 |

Typically, the anchor point within a trust region is selected based on the current best observed value from the accumulated data. However, in our approach, we enhance exploration by sampling anchor points based on their objective values. We apply a softmax function to the objective values of the data points to determine their probabilities of being selected as anchor points. This probability is defined as: $p(\mathbf{x}^{(i)}) = \frac{\exp(y^{(i)}/\tau')}{\sum_j \exp(y^{(j)}/\tau')}$ where $y^{(i)}$ is the objective value of point $i$, and $\tau'$ is the temperature parameter set to 0.1, facilitating a more explorative selection by emphasizing higher objective values. This method ensures that points with higher objective values are more likely to be selected, promoting a diverse exploration of the solution space.

## J  BASELINES

In the Guacamol benchmark, we use the following LBO methods as the baselines:

- LSBO: searches the entire latent space without any modifications.

- TuRBO-$L$ (Eriksson et al., 2019): employs a trust region strategy, focusing the search on promising areas around the current best score.

- W-LBO (Tripp et al., 2020): utilizes weighted retraining to better adapt the model based on promising new data.

- LOLBO (Maus et al., 2022): integrates joint training between the surrogate and generative models to optimize performance.

- CoBO (Lee et al., 2023): uses Lipschitz regularization to enhance the correlation between the latent space and the objective function, aiming to improve the model's predictive alignment with desired outcomes.

- PG-LBO (Chen et al., 2024): applies pseudo-labeling techniques to predict labels of unlabeled data points, potentially uncovering valuable areas of the search space.

## K  ADDITIONAL EXPERIMENTAL RESULTS

**Analysis of SeqFlow for value discrepancy problem.**  We presented an ablation study of our generative model (SeqFlow) to demonstrate the impact of the value discrepancy problem. We compare NF models by applying different mapping functions: Eq. (8), (9) (ours) and BiLSTM (TextFlow (Ziegler & Rush, 2019)). Both models utilize a same Normalizing Flow (NF) framework. However, TextFlow does not ensure the accurate reconstruction of the inputs since it applies BiLSTM to the mapping function. The optimization results are in Table 4. Please note that we do not apply TACS solely to compare generative models. From the table, our SeqFlow model achieves better performance with fewer parameters compared to the baseline model. SeqFlow and TextFlow use the same NF model, but TextFlow includes more components and therefore has more parameters. Although TextFlow has more parameters, our SeqFlow model resolves the value discrepancy problem, resulting in higher optimization performance. This shows that addressing the value discrepancy problem is important in effective Bayesian optimization.

Table 4: Optimization results according to different generative models. Each score represents the mean and standard deviation of 5 independent runs.

| Methods | SeqFlow (Ours) | TextFlow (Ziegler & Rush, 2019) |
|---|---|---|
| BO | NF-BO w/o TACS | NF-BO w/o TACS |
| Base Model | Autoregressive NF | Autoregressive NF |
| Mapping Functions | Eq. (8, 9) | BiLSTM |
| Complete Reconstruction | O | X |
| # Params | 31M | 54M |
| adip | $\mathbf{0.778 \pm 0.016}$ | $0.716 \pm 0.017$ |
| med2 | $\mathbf{0.372 \pm 0.012}$ | $0.347 \pm 0.010$ |

**Measurements of the value discrepancy between actual and latents.** To demonstrate that our SeqFlow effectively addresses the value discrepancy problem, we measure the ratio of instances where $y \neq \hat{y}$, comparing the score of the input data $y$ and the reconstructed data $\hat{y}$. We use top 1,000 data points from 10,000 initial data points across all Guacamol tasks. The experimental results are in Table 5. From the table, our SeqFlow model accurately reconstructs every data point, unlike the TextFlow model, which indicates that our SeqFlow model is appropriate NF model to address the value discrepancy problem.

Table 5: Quantitive measurement of value discrepancy. We measure the ratio of $y \neq \hat{y}$.

| Model | SeqFlow | TextFlow (Ziegler & Rush, 2019) |
|---|---|---|
| adip | **0.000** | 0.548 |
| med2 | **0.000** | 0.609 |
| osmb | **0.000** | 0.630 |
| pdop | **0.000** | 0.502 |
| rano | **0.000** | 0.814 |
| zale | **0.000** | 0.750 |
| valt | **0.000** | 0.001 |

**Exploration abilities of TACS according to the number of initial points.** To verify the exploration abilities and effectiveness of our TACS, we conduct an ablation study of TACS using 1 initial data point and 10,000 initial data points in Table 6. Each experiment is repeated 5 times and we report the average and standard deviation of the results. From the table, NF-BO with TACS consistently shows better optimization results compared to NF-BO without TACS when using 1 initial data point. Moreover, NF-BO with TACS is shown to be robust to the number of initial points by comparing the optimization results between NF-BO w/ TACS (init 1) and NF-BO w/ TACS (init 10K). This suggests that TACS has strong exploration capabilities. Interestingly, in some tasks like Adip, NF-BO w/ TACS (init 1) performs better than NF-BO w/ TACS (init 10K), which highlights the effectiveness of TACS under low-data scenarios. We maintained the TACS temperature at 400, consistent with our main experiments.

**Choice of TACS temperature.** To choose the TACS temperature in our main experiments, we initially conduct a simple search for the TACS temperature on one of the Guacamol tasks within the range [400, 200, 100]. Based on this search, we fix the temperature at 400 for all benchmarks and tasks.

To further demonstrate the robustness of the TACS temperature, we provide a sensitivity analysis in Table 7. This analysis is performed across seven Guacamol tasks, with results averaged over five runs per task and summed. Both the oracle budget and the number of initial data were set to 10,000. From the table, TACS temperatures above 200 consistently show better optimization results compared to not using TACS, highlighting the robustness of our approach to the choice of TACS temperature across all tasks.

Table 6: Performance on low-data scenarios with 1 initial data. Bold values indicate the highest performance among methods with only 1 initial data point. Each score represents the mean and standard deviation of 5 independent runs.

| Method | NF-BO w/o TACS (init 1) | NF-BO w/ TACS (init 1) | NF-BO w/ TACS (init 10K) |
|--------|-------------------------|------------------------|--------------------------|
| adip | $0.765 \pm 0.038$ | $\mathbf{0.818 \pm 0.051}$ | $0.809 \pm 0.059$ |
| med2 | $0.306 \pm 0.014$ | $\mathbf{0.307 \pm 0.027}$ | $0.380 \pm 0.014$ |
| osmb | $0.848 \pm 0.037$ | $\mathbf{0.855 \pm 0.007}$ | $0.897 \pm 0.016$ |
| pdop | $0.564 \pm 0.045$ | $\mathbf{0.623 \pm 0.043}$ | $0.759 \pm 0.023$ |
| rano | $0.846 \pm 0.019$ | $\mathbf{0.848 \pm 0.021}$ | $0.941 \pm 0.006$ |
| valt | $0.198 \pm 0.443$ | $\mathbf{0.786 \pm 0.439}$ | $0.995 \pm 0.004$ |
| zale | $0.586 \pm 0.013$ | $\mathbf{0.589 \pm 0.033}$ | $0.760 \pm 0.012$ |

Table 7: Sensitivity analysis of TACS temperature on 7 Guacamol tasks. Each task's performance is averaged over five trials.

| TACS Temperature $\tau$ | Score sum on 7 Guacamol tasks |
|-------------------------|-------------------------------|
| $\infty$ (w/o TACS) | **5.495** |
| 2000 | **5.495 (+0.000)** |
| 1000 | **5.523 (+0.028)** |
| 400 | **5.544 (+0.049)** |
| 200 | **5.565 (+0.070)** |
| 100 | 5.481 (-0.014) |
| 50 | 5.453 (-0.042) |
| 20 | 5.418 (-0.077) |

**Sensitivity analysis for coefficient $\lambda$ to similarity loss.**    We conduct a sensitivity analysis to evaluate the performance across different $\lambda$ values. These experiments are carried out on seven Guacamol tasks, with the results in Table 8. The table demonstrates that our NF-BO model maintains competitive performance across various $\lambda$ settings. Also, tasks such as osmb and rano show robustness to variations in $\lambda$.

Table 8: Sensitivity of $\lambda$ on model performance. Each score represents the mean and standard deviation of 5 independent runs.

| Coefficient $\lambda$ | 0.1 | 1 | 10 |
|-----------------------|-----|---|-----|
| adip | $0.783 \pm 0.024$ | $\mathbf{0.809 \pm 0.059}$ | $0.771 \pm 0.023$ |
| med2 | $0.366 \pm 0.018$ | $\mathbf{0.380 \pm 0.014}$ | $0.367 \pm 0.012$ |
| osmb | $0.895 \pm 0.004$ | $0.897 \pm 0.016$ | $\mathbf{0.900 \pm 0.009}$ |
| pdop | $0.768 \pm 0.033$ | $0.759 \pm 0.023$ | $\mathbf{0.785 \pm 0.028}$ |
| rano | $0.938 \pm 0.004$ | $\mathbf{0.941 \pm 0.006}$ | $0.940 \pm 0.005$ |
| valt | $0.989 \pm 0.006$ | $\mathbf{0.995 \pm 0.004}$ | $0.975 \pm 0.031$ |
| zale | $0.737 \pm 0.014$ | $\mathbf{0.760 \pm 0.012}$ | $0.752 \pm 0.013$ |

**Illustration of the value discrepancy problem on Guacamol tasks.**    We additionally include illustrations similar to those presented in the main paper (see Figure 1) for five additional Guacamol tasks, which are shown in Figure 14.

## L    DISCUSSION WITH LaMBO AND LaMBO-2

In contrast to our model, where the decoder is formulated as our problem statement's decoder $\mathbf{x} = p_\theta(\mathbf{z})$ (Eq. 6), the decoders in methods like LaMBO (Stanton et al., 2022) and LaMBO-2 (Gruver et al., 2024) (*e.g.*, MAE) operate differently since they utilize the original $\mathbf{x}$ in the decoder to

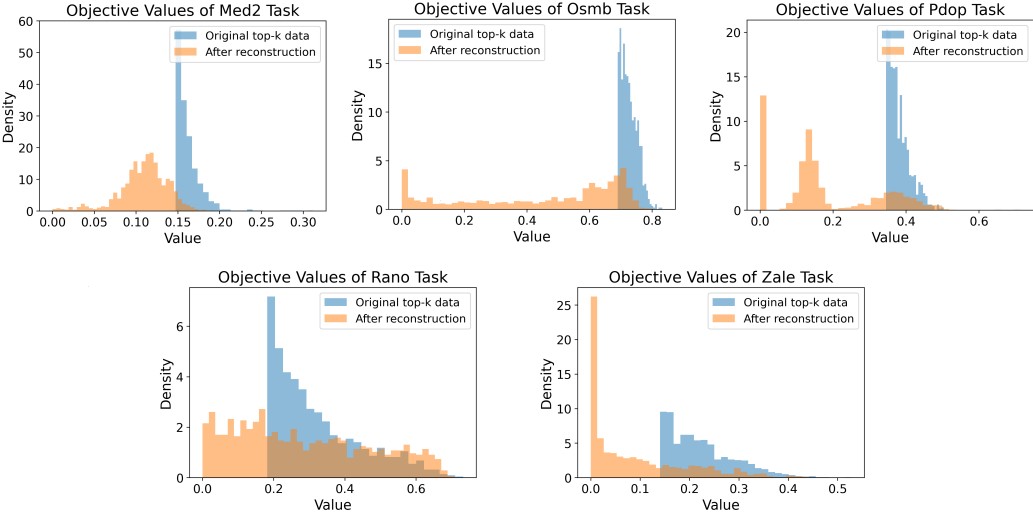

Figure 14: Visualizations of the value discrepancy problem across 5 Guacamol tasks.

restore inputs for unmasked input tokens, thus cannot define the value discrepancy problem, which is defined as Equation (7). Instead, these models elegantly align the latent space and input space by repeatedly decoding or encoding new candidates. One key difference between LaMBO, LaMBO-2 and ours is that our method does not need additional encoding and decoding of all candidates. Our method decodes only the latent vector chosen by the acquisition function thanks to the one-to-one function from the input space to the latent space and its left-inverse function.

## M  GLOSSARY OF NOTATION

Table 9: Glossary of notation.

| | |
|---|---|
| $\mathbf{x}$ | Sequence of discrete data, each $\mathbf{x}_i$ is a token index from $\mathcal{V}$, where $\mathbf{x} \in \mathbb{N}^L$ |
| $\mathbf{v}$ | Continuous representation corresponding to discrete data $\mathbf{x}$, where $\mathbf{v} \in \mathbb{R}^{L \times F}$ |
| $L$ | Number of tokens in a sequence |
| $F$ | Dimension of the embedding space |
| $\mathbf{e}_j$ | Embedding vector of the $j$-th token, where $\mathbf{e}_j \in \mathbb{R}^F$ |
| $\mathrm{sim}(\cdot, \cdot)$ | Cosine similarity function used to measure the similarity between two vectors |
| $a(\mathbf{v}_i)$ | Function that returns the index of the most similar embedding vector to $\mathbf{v}_i$ |
| $p(\mathbf{x}_i \mid \mathbf{v}_i)$ | Conditional probability of the token index $\mathbf{x}_i$ given the continuous vector $\mathbf{v}_i$ |
| $\delta_{\mathbf{x}_i, a(\mathbf{v}_i)}$ | Equals 1 if $\mathbf{x}_i$ is the index returned by $a(\mathbf{v}_i)$ and 0 otherwise |
| $\mathbf{x}^*$ | Optimal value of the optimization |
| $f$ | Objective function |
| $y$ | Objective value of input $\mathbf{x}$ |
| $\mathbf{z}$ | Latent vector of input $\mathbf{x}$ |
| $\tilde{\mathbf{x}}, \tilde{y}, \tilde{\mathbf{z}}$ | Next query data selected by the acquisition function |
| $g$ | Flow transformation |
| $\mathbf{g}$ | Sequence of flow transformations |
| $\omega_i(z)$ | Pointwise mutual information of $\mathbf{x}$ and $\mathbf{z}_i$ |
| $\pi_i(z)$ | Token-level sampling probability on the latent $\mathbf{z}$ |
| $\kappa$ | Constant scaling factor varied by sequence length |
| $\tau$ | Temperature parameter for Token-level Adaptive Candidate Sampling (TACS) |

