# OpenReview forum: "Latent Bayesian Optimization via Autoregressive Normalizing Flows"
_ICLR.cc/2025/Conference — ICLR 2025 Oral_

### Official Review · Reviewer_TFzn · 2024-10-27

**Soundness:** 4
**Presentation:** 3
**Contribution:** 4
**Rating:** 8
**Confidence:** 3

**Summary:**

This paper introduces a Latent Bayesian Optimization (L-BO) method, called NF-BO, that aims to improve BO performances by avoiding the value discrepancy problem typical of existing L-BO methods.
This method uses a sequential normalizing flow-based model to ensure a 1 to 1 injective mapping between latents and input data points, therefore avoiding the value-discrepancy problem. The authors also introduce TACS, an adaptive sampling methods for candidate points to increase the efficiency of the search by better focusing on areas with higher potential impact.
The effectiveness of NF-BO is demonstrated on a wide range of molecule generation tasks.

**Strengths:**

1. Efficient BO methods are key to finding optimal solution to a wide range of complicated problems with expensive and black-box objective functions. Many of these problems have inherently discrete components, and as such it's key to develop methods like NF-BO that can effectively deal with discrete spaces in a principled manner.
2. The authors give a good overview of the limitations of current methods, and the reason why normalizing flows are a better choice for this task.
3. To the best of my knowledge, this method is novel in the usage of NF, the SeqFlow architecture and the TACS sampling.
4. In the experimental benchmarks NF-BO  together with TACS sampling greatly outperforms competing methods, leading to much better solutions in a fraction of the oracle calls

**Weaknesses:**

1. Overall, I found some of the SeqFlow explanations in section 4.2 a bit too abstract to be easily understood by the reader. I think the section could benefit from using a simple running example to show in practice (a) how an embedding dictionary could look like (b) a sample surrogate model and acquisition function
2. In the paper you claim that current L-BO methods fail because of the value discrepancy problem. Are you sure that the issues are due to the value discrepancy problem itself, and not for example due to a "poor" learned latent space? In other words, is "perfect reconstruction" a must, or "good enough reconstruction" would also work?
3. How sensitive is the method to different values of $\lambda$  in the similarity loss in (14)?

**Questions:**

See weaknesses section.

---

> ### Author Response · Authors · 2024-11-22
> **Response to Reviewer TFzn**
>
> **[W1] More explanation of Section 4.2.**
>
> Thank you for the suggestion. For a better understanding of our NF-BO, we modify Figure 4 in the main paper to show the details of the embedding dictionary. In detail, the latent embedding (embedding dictionary) $\mathbf{e} \in \mathbb{R}^{\left\lvert \mathcal{V}\right\rvert \times F}$ consists of $\left\lvert \mathcal{V}\right\rvert$ embedding vectors $\boldsymbol{e}_j \in \mathbb{R}^F, j \in 1, 2, \dots \left\lvert \mathcal{V}\right\rvert$. Each embedding vector $\mathbf{e}_j$ corresponds to the $j$-th alphabet in the vocabulary set $\mathcal{V}$. These embedding vectors are initialized with a normal distribution. For the surrogate model, we use a sparse variational Gaussian process [1] enhanced with a deep kernel [2].
>
> **[W2] Impacts of the value discrepancy problem on NF-BO.**
>
> We additionally conduct an ablation study of our generative model (SeqFlow) to demonstrate the impact of the value discrepancy problem. We compare generative models by applying different mapping functions to the same normalizing flow framework: Equation (8, 9) (ours) and BiLSTM (TextFlow [8]). TextFlow [8] does not ensure the accurate reconstruction of the inputs since it applies BiLSTM to the mapping function. The optimization results are in Table 1. Please note that we do not apply TACS solely to compare generative models. From the table, our SeqFlow model achieves better performance with fewer parameters compared to the baseline model. This shows that addressing the value discrepancy problem is important in effective Bayesian optimization.
>
> **Table 1: Optimization performance on two Guacamol tasks.**
>
> | Method | **SeqFlow (Ours)** | **TextFlow [3]**  | **SELFIES VAE** |
> | --- | --- | --- | --- |
> | **BO** | **NF-BO w/o TACS** | **NF-BO w/o TACS** | **LOL-BO** |
> | **Base Model** | **Autoregressive NF** | **Autoregressive NF**  | **SELFIES VAE** |
> | **Mapping Functions** | **Eq. (8, 9)** | **BiLSTM** |  |
> | **Complete Reconstruction** | **O** | **X** | **X** |
> | adip | **0.778** $\pm$ **0.016** | 0.716 $\pm$ 0.017 | 0.716 $\pm$ 0.003 |
> | med2 | **0.372** $\pm$ **0.012** | 0.347 $\pm$ 0.010 | 0.325 $\pm$ 0.004 |
>
> Furthermore, to demonstrate that our SeqFlow effectively addresses the value discrepancy problem, we measure the ratio of instances where $y\ne\hat{y}$, comparing the score of the input data $y$ and the reconstructed data $\hat{y}$. We use top 1,000 data points from 10,000 initial data points across all Guacamol tasks. The experimental results are in Table 2. From the table, our SeqFlow model accurately reconstructs every data point, unlike the TextFlow model, which indicates that our SeqFlow model is appropriate NF model to address the value discrepancy problem.
>
> Moreover, we add visualizations of the value discrepancy problem on other Gucamol Tasks in the appendix to show that the value discrepancy problem generally arises.
>
> **Table 2: Quantitive measurement of value discrepancy using the ratio of** $y \ne \hat{y}$
>
> | Model | SeqFlow | TextFlow [8] |
> | --- | --- | --- |
> | adip | 0.000 | 0.548 |
> | med2 | 0.000 | 0.609 |
> | osmb | 0.000 | 0.630 |
> | pdop | 0.000 | 0.502 |
> | rano | 0.000 | 0.814 |
> | zale | 0.000 | 0.750 |
> | valt | 0.000 | 0.001 |
>
>
> **[W3] Sensitivity Analysis for $\lambda$.**
>
> We note that the value of $\lambda=1$ was fixed in all experiments in the main paper without searching. As suggested, we conduct a sensitivity analysis to evaluate the performance across different $\lambda$ values. The experiments are carried out on seven Guacamol tasks, with the results in Table 4. The table demonstrates that our NF-BO model maintains competitive performance across various $\lambda$ settings. Also, tasks such as osmb and rano show robustness to variations in $\lambda$.
>
> **Table 4: Sensitivity of $\lambda$ on Model Performance.**
>
> | $\lambda$ | 0.1 | 1 | 10 |
> | --- | --- | --- | --- |
> | adip | 0.783 + 0.024 | **0.809 + 0.059** | 0.771 + 0.023 |
> | med2 | 0.366 + 0.018 | **0.380 + 0.014** | 0.367 + 0.012 |
> | osmb | 0.895 + 0.004 | 0.897 + 0.016 | **0.900 + 0.009** |
> | pdop | 0.768 + 0.033 | 0.759 + 0.023 | **0.785 + 0.028** |
> | rano | 0.938 + 0.004 | **0.941 + 0.006** | 0.940 + 0.005 |
> | valt | 0.989 + 0.006 | **0.995 + 0.004** | 0.975 + 0.031 |
> | zale | 0.737 + 0.014 | **0.760 + 0.012** | 0.752 + 0.013 |
>
> [1] Snelson, Edward, and Zoubin Ghahramani. "Sparse Gaussian processes using pseudo-inputs." NeurIPS 2005.
>
> [2] Wilson, Andrew Gordon, et al. "Deep kernel learning." AISTATS 2016.
>
> [3] Ziegler, Zachary, and Alexander Rush. "Latent normalizing flows for discrete sequences." ICML 2019.

---

> > ### Comment · Reviewer_TFzn · 2024-11-23
> > **Confirm my score**
> >
> > Thanks for your answer. I will confirm my score and argue for acceptance

---

### Official Review · Reviewer_xNso · 2024-10-30

**Soundness:** 3
**Presentation:** 3
**Contribution:** 4
**Rating:** 8
**Confidence:** 5

**Summary:**

&nbsp;

The authors present the first architecture featuring a normalizing flow as the encoder-decoder model for latent space Bayesian optimization. The invertibility of the normalizing flow is important in addressing the value discrepancy problem, namely that encoded points are not guaranteed to be reconstructed perfectly by the decoder. This pathology in latent space Bayesian optimization has been identified in recent work [11] under the name of the "misalignment problem". It should be noted that [11], to the best of my knowledge is not yet publicly available, and hence the authors should not be required to consider this work. Furthermore, the authors introduce a token-level adaptive sampling scheme (TACS) and demonstrate that it improves performance of the NF-BO method in an ablation study. I think the method is innovative and the empirical results are impressive. However, I have major concerns about the reproducibility of the results, given that the codebase is not provided. I believe crucial details to reproduce the experiments are missing as detailed below. If these issues can be addressed during the rebuttal phase I will be more than happy to increase my score and recommend acceptance.

&nbsp;

**Strengths:**

&nbsp;

1. The work is the first to introduce normalizing flows as an encoder-decoder model in latent space Bayesian optimization. This represents a novel departure to existing work.

2. The authors methodology addresses an important pathology in latent space Bayesian optimization by enforcing invertibility on the encoder-decoder architecture.

3. The authors demonstrate strong empirical results and obtain state-of-the-art performance on a widely-used benchmark (PMO) where they demonstrate that NF-BO outperforms a suite of other optimization approaches including RL, genetic algorithms, and other Bayesian optimization methods. As such, the current work is of very broad interest to the optimization community.

&nbsp;

**Weaknesses:**

&nbsp;

**MAJOR POINTS**

&nbsp;

1. My principal concerns relate to the reproducibility of the empirical results since the authors do not provide their code. An Anonymous GitHub link or similar could be provided during the rebuttal phase. In its current form, I don't believe the results are reproducible from the manuscript alone since crucial details such as the GP surrogate model (vanilla GP/sparse GP) and the choice of acquisition function (EI, PI, Thompson sampling) appear to be missing.

2. I experienced difficulty navigating Section 4.2 as the notation is very confusing. On line 239, what is L, the number of tokens in a sequence? If so why is each token a vector? In the case of SMILES or SELFIES strings of molecules each token will be an alphabetic character which should not be a vector quantity? What is F? The dimension of the embedding space? What is Equation 8 implying? In other words what does choosing the j that maximizes sim(v_i, e_j) entail? The argmax variable is not clear as only the index is given. A similar point is applicable for the definition of a_i(v). How are the embeddings e computed? Perhaps a glossary of notation would be beneficial for clarity?

&nbsp;

**MINOR POINTS**

&nbsp;

1. In the abstract, with reference to the sentence, "accurate and one-to-one mappings between latent and input spaces". Is the term "accurate" redundant since a one-to-one mapping implies perfect reconstruction?

2. When introducing Bayesian optimization, it may be worth citing the originating papers for the method [1, 2] as discussed in [3].

3. In the first paragraph, some citations are not parenthetical \citet in LateX vs. \cite and should be parenthetical.

4. The sentence on line 50, "This makes the value discrepancy problem more and makes
the searching challenging." Should be revised.

5. In the related work section on latent space Bayesian optimization, it would also be worth discussing the following papers [4-8] and their relationship to the current work e.g. [8] explicitly considers the diversity in the candidate solutions in a similar fashion to the Token-level Adaptive Candidate Sampling (TACS) employed by the authors.

6. There are some missing capitalizations in the references e.g. "Bayes" instead of "bayes" in the citation for Kingma and Welling, 2014.

7. On line 147, it would be worth changing the notation for the label y to be a scalar since f is defined as a scalar quantity.

8. Missing full stop at the end of Equation 4.

9. The decision to use the variable v in Equation 3 is slightly confusing. Why not just use x?

10. The problem formulation of latent space Bayesian optimization in Equation 5, differs from that in e.g. [9] where the goal is to identify z^* that maximizes the expected f(p(z)) under a probabilistic decoder. I'm assuming that the formulation presented assumes a deterministic decoder since the normalizing flow is capable of producing a bijective mapping. It may hence be worth emphasizing the distinction between the problem formulation with normalizing flows (deterministic decoder) and the problem formulation with VAEs (probabilistic decoder).

11. Typo line 200, y^(i) should be a scalar, similarly line 201.

12. For the contrastive loss in Equation 13, it would be good to discuss the intent in relation to deep metric learning in [9]. The goal of the contrastive loss would appear to be to structure the latent space. As found in [9] it might be an idea to leverage label information y to structure the normalizing flow latent space. This could yield improvements to the GP fit.

13. In Table 1, what is the score? The sum of the scores of each individual task constituting the PMO benchmark? It would be worth specifying this directly in the main paper in addition to the appendix (I only found this information later when I read the appendix.). Additionally, it would be worth mentioning that the full breakdown of results from the PMO benchmark [10] is provided in the appendix.

14. Typo line 484, the number of the referenced table is missing.

15. Typo line 485, "metrics".

16. In Figures 5 and 7, the method for computing the errorbars should be reported e.g. is the standard error or the standard deviation over 5 trials being depicted? It would also be worth stating the number of trials directly in the figure captions.

17. Typo line 893, "number of query points".

&nbsp;

**REFERENCES**

&nbsp;

[1] Kushner, HJ., [A Versatile Stochastic Model of a Function of Unknown and Time
Varying Form](https://www.sciencedirect.com/science/article/pii/0022247X62900112). Journal of Mathematical Analysis and Applications 5(1):150–167. 1962.

[2] Kushner HJ., [A New Method of Locating the Maximum Point of an Arbitrary Multipeak Curve in the Presence of Noise](https://asmedigitalcollection.asme.org/fluidsengineering/article-abstract/86/1/97/392213/A-New-Method-of-Locating-the-Maximum-Point-of-an?redirectedFrom=fulltext). Journal of Basic Engineering 86(1):97–106. 1964.

[3] Garnett, R., [Bayesian optimization](https://bayesoptbook.com/). Cambridge University Press. 2023.

[4] Verma, Ekansh, Souradip Chakraborty, and Ryan-Rhys Griffiths. [High dimensional Bayesian optimization with invariance.](https://realworldml.github.io/files/cr/paper53.pdf) In ICML Workshop on Adaptive Experimental Design and Active Learning. 2022.

[5] Stanton, S., Maddox, W., Gruver, N., Maffettone, P., Delaney, E., Greenside, P. and Wilson, A.G., 2022, June. [Accelerating Bayesian optimization for biological sequence design with denoising autoencoders.](https://proceedings.mlr.press/v162/stanton22a/stanton22a.pdf) In International Conference on Machine Learning (pp. 20459-20478). PMLR.

[6] Notin, P., Hernández-Lobato, J.M. and Gal, Y., 2021. [Improving black-box optimization in VAE latent space using decoder uncertainty.](https://openreview.net/pdf?id=F7LYy9FnK2x) Advances in Neural Information Processing Systems, 34, pp.802-814.

[7] Lu, X., Gonzalez, J., Dai, Z. and Lawrence, N.D., 2018, [Structured variationally auto-encoded optimization.](https://proceedings.mlr.press/v80/lu18c.html) In International Conference on Machine Learning (pp. 3267-3275). PMLR.

[8] Maus, N., Wu, K., Eriksson, D. and Gardner, J., 2023, [Discovering Many Diverse Solutions with Bayesian Optimization.](https://proceedings.mlr.press/v206/maus23a/maus23a.pdf) In International Conference on Artificial Intelligence and Statistics (pp. 1779-1798). PMLR.

[9] Grosnit, Antoine, Rasul Tutunov, Alexandre Max Maraval, Ryan-Rhys Griffiths, Alexander I. Cowen-Rivers, Lin Yang, Lin Zhu et al. [High-dimensional Bayesian optimisation with variational autoencoders and deep metric learning.](https://arxiv.org/abs/2106.03609) arXiv preprint arXiv:2106.03609. 2021.

[10] Gao, Wenhao, Tianfan Fu, Jimeng Sun, and Connor Coley. [Sample efficiency matters: a benchmark for practical molecular optimization.](https://proceedings.neurips.cc/paper_files/paper/2022/hash/8644353f7d307baaf29bc1e56fe8e0ec-Abstract-Datasets_and_Benchmarks.html) Advances in Neural Information Processing Systems 35, 21342-21357. 2022.

[11] Chu et al. [Inversion-Based Latent Bayesian Optimization](https://nips.cc/virtual/2024/poster/95013), NeurIPS 2024.

&nbsp;

**Questions:**

&nbsp;

1. What is the relationship between the similarity loss in Equation 13 and deep metric learning [9]? Do the authors think that incorporating label information y into the similarity loss could yield a better surrogate fit as in [9]? Did the authors consider incorporating label information into the loss formulation?

2. For Equation 14, did the authors consider coordinate ascent on the two terms in the objective as opposed to scalarization with the lambda parameter? Coordinate ascent may be more efficient and would avoid setting an arbitrary scaling parameter.

3. On line 331, the authors state, "In practice, sampling the full posterior function from the GP posterior distribution is infeasible". What do the authors mean by this and why is it relevant. Do the authors assume an acquisition function like Thompson Sampling is being used?

4. In Appendix C, do the authors have any intuition for why lower temperature yields reduced diversity for the rano task?

5. In Appendix F, how is the embedding dictionary specified?

6. In Algorithm 1, what is topK on line 2?

&nbsp;

**Details Of Ethics Concerns:**

&nbsp;

No ethical concerns.

&nbsp;

---

> ### Author Response · Authors · 2024-11-22
> **Response to Reviewer xNso (1/2)**
>
> We really appreciated Reviewer XNso for the thoughtful comments and feedback. We address the raised concerns below.
>
> **[W1] Implementation details and codes.**
>
> For reproducibility, we provide an anonymous GitHub repository [[link](https://anonymous.4open.science/r/NFBO-EA7C/)], which includes instructions for running Guacamol experiments. Additionally, we added the details of our NF-BO in the Section 5.3, as suggested. Specifically, we employed Thompson sampling [1] as the acquisition function, and use a sparse variational Gaussian process [2] enhanced with a deep kernel [3] as the surrogate model.
>
> **[W2] Clear Notations in Section 4.2.**
>
> Sorry for the confusion. As suggested, we add the glossary of notation in the Appendix M. Additionally, we update Figure 4 to better illustrate the NF-BO process, including examples. The sequence $\mathbf{x} = \left[\mathbf{x}_1, \dots, \mathbf{x}_L \right] \in \mathcal{X}$ consists of $L$ number of tokens (e.g., C, H). Each token (alphabet) has its corresponding latent vector with the dimension of $F$. As shown in Figure 4, Equation (8) plays a role in converting the continuous representation $\mathbf{v}$ into the discrete token sequence during the decoding phase. It selects tokens by comparing the representation with each embedding vector $\mathbf{e}_j$ in the embedding’s codebook in the decoding process. These embedding vectors are initialized with a normal distribution.
>
> **[W3] Typos and clearness (Minor points 1, 4, 6, 7, 8, 11, 14, 15, 17.)**
>
> We appreciate the detailed feedback and have addressed these points by revising the main paper and appendix as suggested.
>
> **[W4] Citations (Minor points 2, 3, 5)**
>
> Thank you for the suggestions. We add citations for the originating papers and update the non-parenthetical citations as recommended.
>
> **[W5] Use the variable $\mathbf{x}$ instead of $\mathbf{v}$ in Equation 3. (Minor points 9)**
>
> We use $\mathbf{v}$ instead of the variable $\mathbf{x}$, since the input of normalizing flow model needs to be continuous.
>
> **[W6] Clarification on decoder assumptions in LBO Formulation (minor points 10)**
>
> Thank you for the feedback. In our LBO formulation, as presented in Equation 5, we assume a deterministic decoder. In contrast, stochastic decoders are typically designed to maximize the expectation of $f(p_\theta(\mathbf{z}))$. We will clarify in the main paper.
>
> **[W7] Details of the “score” in Table 1. (Minor points 13).**
>
> Thank you for the feedback. We add a detailed description of the score used in the PMO benchmarks to both the main text and the caption of Table 1.
>
> **[W8] Full results on PMO benchmark (Minor points 13).**
>
> Thank you for the suggestion. As suggested, we add the mention in the main text that the full results can be found in the appendix.
>
> **[W9] Details of computing the errorbars (Minor points 16)**
>
> Thank you for the feedback. We add that the range indicates the standard error and the number of trials in the figure caption.

---

> > ### Comment · Reviewer_xNso · 2024-11-24
> > **Thank you for the Update**
> >
> > &nbsp;
> >
> > Many thanks to the authors for their updates.
> >
> > &nbsp;
> >
> > **[W1] Implementation details and Code.**
> >
> > I believe the open-sourcing of the codebase has significantly strengthened the work. One point to note is the citation for the sparse variational GP. While Snelson et al. did indeed introduce inducing point GPs, the GPyTorch documentation for the `VariationalStrategy` class (https://github.com/cornellius-gp/gpytorch/blob/main/gpytorch/variational/variational_strategy.py) cites the references [1,2] for the implementation.
> >
> > &nbsp;
> >
> > **[W2-9] Clarity.**
> >
> > Many thanks to the authors for making the suggested changes.
> >
> > &nbsp;
> >
> > **__REFERENCES__**
> >
> > &nbsp;
> >
> > [1] Hensman, J., Matthews, A. and Ghahramani, Z., [Scalable variational Gaussian process classification.](http://proceedings.mlr.press/v38/hensman15.pdf) In Artificial Intelligence and Statistics (pp. 351-360). PMLR. 2015.
> >
> > [2] Matthews, A. G. D. G. (2017). [Scalable Gaussian process inference using variational methods](https://www.repository.cam.ac.uk/items/a2c7e9a7-d7bb-40c0-9285-8e0f487869e1) [Apollo - University of Cambridge Repository]. https://doi.org/10.17863/CAM.25348
> >
> > &nbsp;

---

> ### Author Response · Authors · 2024-11-22
> **Response to Reviewer xNso (2/2)**
>
> **[Q1] Similarity Loss with Label y**
>
> We introduced a similarity loss for training the embedding vector $\mathbf{e}_i$ rather than the latent space $\mathbf{z}$. Thus, incorporating labels directly into the training of each embedding vector proves challenging. Instead, as suggested, utilizing label $y$ to further train the latent space could be effective in learning a better surrogate model.
>
> **[Q2] Coordinate ascent for tuning $\lambda$.**
>
> In hyperparameter tuning, using a first-order method can lead to an unbounded solution, such as negative infinity. Approaches like bi-level optimization, Meta-learning, or autoML methods could potentially address this issue. However, applying these methods falls outside the scope of our current study.
>
> **[Q3] What is “Sampling the full posterior function from the GP posterior distribution is infeasible?” and why state it?**
>
> We apologize for the confusion regarding our statement. We write the sentence "Sampling the full posterior function from the GP posterior distribution is infeasible."  to clarify our use of Thompson sampling on "a finite candidate set". As said, our TACS assumes utilizing Thompson sampling. We add the elaboration of it in our main paper.
>
> **[Q4] Have any intuition about RANO task in Figure 11?**
>
> We believe that the RANO task requires different temperature ranges to effectively promote exploration. We hypothesize that the average length of the input token sequence or the distribution of high-scoring samples may influence these differing trends. To investigate these hypotheses, we are currently designing additional experiments and will provide updates as soon as the results are available.
>
> **[Q5]** **How is the Embedding Dictionary Specified?**
>
> The latent embedding (embedding dictionary) $\mathbf{e} \in \mathbb{R}^{\left\lvert \mathcal{V}\right\rvert \times F}$ consists of $\left\lvert \mathcal{V}\right\rvert$ embedding vectors $\boldsymbol{e}_j \in \mathbb{R}^F, j \in 1, 2, \dots \left\lvert \mathcal{V}\right\rvert$. Each embedding vector $\mathbf{e}_j$ corresponds to the $j$-th alphabet in the vocabulary set $\mathcal{V}$. These embedding vectors are initialized with a normal distribution. We add further details of the embedding in our main paper and Figure 4.
>
> **[Q6] Details of topK on Line 2 in Algorithm 1.**
>
> We add the details of topK in our manuscript: topK in the algorithm refers to selecting the top $k$ data points with the highest objective values from the dataset $\mathcal{D}$. The number of data $k$ is specified in Table 3 of the appendix.
>
> [1] Eriksson, David, et al. "Scalable global optimization via local Bayesian optimization." NeurIPS 2019.
> [2] Snelson, Edward, and Zoubin Ghahramani. "Sparse Gaussian processes using pseudo-inputs." NeurIPS 2005.
> [3] Wilson, Andrew Gordon, et al. "Deep kernel learning." AISTATS 2016.

---

> > ### Comment · Reviewer_xNso · 2024-11-25
> > **Comments on Updated Manuscript**
> >
> > &nbsp;
> >
> > Many thanks to the authors for clarifying my questions. With reference to the updated manuscript, in terms of the statement,
> >
> > &nbsp;
> >
> > __CoBO (Lee et al., 2023) implements a novel loss function to enhance the alignment between the latent space and the objective function. However, these region-based methods still encounter the value discrepancy problem, where the output value from the decoded input is inconsistent with the original value.__
> >
> > &nbsp;
> >
> > Both CoBO and LOL-BO (verifiable via the open-source implementation) implement a re-centering technique which seeks to address the value discrepancy problem, albeit they incur additional oracle calls to do so. It would be worth discussing the merits of recentering viz-a-viz the authors' method in the updated manuscript.
> >
> > &nbsp;

---

> > > ### Comment · Reviewer_xNso · 2024-11-25
> > > **Upgrading Score**
> > >
> > > &nbsp;
> > >
> > > I am upgrading my score to recommend acceptance based on the code supplied by the authors. I look forward to hearing further updates on the progress of the RANO experiments.
> > >
> > > &nbsp;

---

> > > > ### Author Response · Authors · 2024-11-28
> > > > **Response to Reviewer xNso1 (1/2)**
> > > >
> > > > We appreciate Reviewer xNso for acknowledging the strengths of our paper and providing constructive feedback. We happily provide the response to the following questions below:
> > > >
> > > > **[1] Additional citation.**
> > > >
> > > > Thanks for the feedback. We added the suggested citations to our paper at L430.
> > > >
> > > > **[2] Discussion of “Recentering” [1] with the value discrepancy problem.**
> > > >
> > > > Thanks for the comment. The “value discrepancy problem” is defined as Eq. (7):
> > > >
> > > > $$
> > > > \mathbf{x} \neq \hat{\mathbf{x}} \text{ where } \hat{\mathbf{x}}:=p_\theta(q_\phi(\mathbf{x})), \text{and } y \neq \hat{y}.
> > > > $$
> > > >
> > > > This leads to two issues:
> > > >
> > > > - **Inaccurate local search:** In standard Bayesian optimization, local searches operate around the current solution $\mathbf{x}$. In Latent Bayesian Optimization (LBO), these searches should ideally operate around the latent $\mathbf{z}$ that generates $\mathbf{x}$. Due to the value discrepancy problem, $\mathbf{z}$ generates $\hat{\mathbf{x}}=p_\theta(\mathbf{z})$ instead of $\mathbf{x}$, causing the search to perform around $\hat{\mathbf{x}}$ rather than the true solution $\mathbf{x}$. This leads to suboptimal search outcomes.
> > > > - **Misalignment problem [2]:** The value discrepancy problem also leads to a mismatch between the outcome of the generated data $\hat{y}$ and original data $y$, where $y\ne \hat{y}$. This mismatch avoids generating the properly aligned data $\left(\mathbf{x}, \mathbf{z}, y \right)$, where $\mathbf{x} = p_{\theta}(\mathbf{z} )$ is generated by the decoder $p_{\theta}: \mathcal{Z} \mapsto \mathcal{X}$ and $y=f(\mathbf{x} )$ is the objective value of $\mathbf{x}$ obtained from the objective function $f: \mathcal{X} \mapsto \mathcal{Y}$. This hinders the learning of an accurate **surrogate model** $g$, which is critical for effective optimization. This issue is referred to as the "misalignment problem" [2].
> > > >
> > > > The **“Recentering” approach used in LOL-BO and CoBO** addresses the “misalignment problem” [2] by requesting additional oracle calls $\hat{y}=f(\hat{\mathbf{x}})$ to generate the aligned data $\left(\hat{\mathbf{x}}, {\mathbf{z}}, f(\hat{\mathbf{x}}) \right)$, where $\hat{\mathbf{x}} = p_\theta(\mathbf{z})$.
> > > >
> > > > However, recentering does not fully resolve the value discrepancy problem. Even after recentering, the latent $\mathbf{z}$ still fails to reconstruct the original data $\mathbf{x}$ (*i.e*., $p_\theta(\mathbf{z}) = \hat{\mathbf{x}} \neq \mathbf{x}$), accurately. This discrepancy continues to induce an inaccurate local search.
> > > >
> > > > In contrast, NF-BO eliminates the value discrepancy problem through its one-to-one mapping property, which ensures that $\mathbf{x} = \hat{\mathbf{x}}$ and $y = \hat{y}$. By directly aligning data points using encoding alone, without requiring additional oracle calls, NF-BO resolves both the value discrepancy and misalignment issues effectively.

---

> > > > > ### Author Response · Authors · 2024-11-28
> > > > > **Response to Reviewer xNso1 (2/2)**
> > > > >
> > > > > **[3] Intuition behind Figure 11 for the RANO Task.**
> > > > >
> > > > > > We hypothesize that the average length of the input token sequence or the distribution of high-scoring samples may influence these differing trends.
> > > > > >
> > > > >
> > > > > As mentioned in our previous response, we analyze the relationship between sequence length and objective value across Guacamol tasks. Additionally, we measure the PMI value $\omega_i$ in Equation (17), which is used to compute the token-level sampling probability $\pi_i(\mathbf{z})$ as defined in Equation (18):
> > > > >
> > > > > $$
> > > > > \pi_i(\mathbf{z} )=\min (0.1 \cdot L\cdot s_i(\mathbf{z} ), 1 ), \quad s_i(\mathbf{z} )=\frac{\exp\left(\omega_i(\mathbf{z})/\tau \right)}{\sum_j \exp\left(\omega_j(\mathbf{z}\right)/\tau )}.
> > > > > $$
> > > > >
> > > > > Table 1 shows **(1)** the correlation between sequence length $L$ and objective value $y$, **(2)** the average sequence length $L$ of the top-$k$ data with the highest objective values, **(3)** the average PMI value $\omega_i$.
> > > > >
> > > > > The results show that RANO task has the highest correlation between sequence length and objective value among all tasks, which indicates that long sequences dominate top-$k$ data points. Additionally, RANO also has the highest PMI values, as the PMI value $\omega_i$ is influenced by the potential variations in the input sequence. So, longer sequences naturally increase the combinatorial diversity of token arrangements, thereby amplifying the PMI values.
> > > > >
> > > > > These findings align with Figure 11, which shows that the RANO task has the highest ratio of distinct samples without TACS (96%) across all tasks. This suggests that tasks like RANO, which require identifying longer sequences, demand a higher temperature $\tau$ to reduce the influence of PMI score $\omega_i$ and ensure sampling probaiblities $\pi$ remain consistent with those of other tasks.
> > > > >
> > > > > **Table 1: Analysis of sequence length and PMI value $\omega_i$ on Guacamol tasks.**
> > > > >
> > > > > | Tasks | $Corr(L, y)$ | $Avg(L)$ (top 1%) | $Avg(\omega_i)$ |
> > > > > | --- | --- | --- | --- |
> > > > > | adip | -0.1401 | 44.35 | 9.0332 |
> > > > > | med2 | 0.2544 | 52.85 | 10.8674 |
> > > > > | osmb | 0.2041 | 50.10 | 10.1218 |
> > > > > | pdop | 0.0557 | 62.08 | 11.3104 |
> > > > > | **rano** | **0.5083** | **75.11** | **21.8375** |
> > > > > | zale | -0.1741 | 40.43 | 7.8500 |
> > > > > | valt | 0.0181 | 43.71 | 8.7982 |
> > > > >
> > > > > [1] Maus, Natalie, et al. "Local latent space bayesian optimization over structured inputs." NeurIPS 2022.
> > > > >
> > > > > [2] Chu, Jaewon, et al. "Inversion-based Latent Bayesian Optimization." arXiv 2024.

---

> > > > > ### Comment · Reviewer_xNso · 2024-11-29
> > > > >
> > > > > &nbsp;
> > > > >
> > > > > **[2] Discussion of Recentering**
> > > > >
> > > > > &nsbp;
> > > > >
> > > > > It would be worth the authors adding the discussion of the recentering, value discrepancy, and misalignment [1] problems to the updated manuscript.
> > > > >
> > > > > &nbsp;
> > > > >
> > > > > **[3] RANO task**
> > > > >
> > > > > &nsbp;
> > > > >
> > > > > Many thanks additionally, for the additional diagnostic analysis on the RANO task, the correlation between sequence length and objective function value appears to be a convincing explanation.
> > > > >
> > > > > &nbsp;
> > > > >
> > > > > **__REFERENCES__**
> > > > >
> > > > > &nbsp;
> > > > >
> > > > > [1] Chu, J., Park, J., Lee, S. and Kim, H.J., 2024. [Inversion-based Latent Bayesian Optimization.](https://arxiv.org/abs/2411.05330) arXiv preprint arXiv:2411.05330.
> > > > >
> > > > > &nbsp;

---

### Official Review · Reviewer_ajp3 · 2024-11-02

**Soundness:** 4
**Presentation:** 4
**Contribution:** 4
**Rating:** 8
**Confidence:** 4

**Summary:**

The authors focus on latent-space Bayesian Optimization (LBO), which in recent years has emerged as an important area of research. The authors' main contribution is two-fold. (i) To remove the discrepancy between the encoder and decoder of the VAE models commonly used in LBO approaches, they learn a normalizing-flow model that preserves a one-to-one mapping and thus a loss-less mapping from observed to encoded space. (ii) an improved approach to sample new candidates from the latent space based on their relative importance.
The method is evaluated on a range of benchmarks with strong performance improvements upon the baselines.

**Strengths:**

- The paper is well-written and the proposal is well-motivated.
- The evaluation is extensive and demonstrates a strong improvement
- Each of the parts, the autoregressive flow as well as the sampler seem straightforward to implement and use for further applications

**Weaknesses:**

The weaknesses listed below are only minor.

- Unless I am mistaken, the final acquisition function used is never specified in the text or appendix.
- In l255 the relation to Ho et al. (2019) remains unclear in the text
- The embedding e is never fully specified.

**Questions:**

- Q1: The paper contains the ablation of running the method with and without TACS, showing a clear improvement when including it. Can the authors speculate (or provide if time permits) on the expected results of a second ablation that combines TACS with a VAE model instead of a flow? Would its relative performance improve to a similar extent as in the proposed approach, or more/less?

---

> ### Author Response · Authors · 2024-11-22
> **Response to Reviewer ajp3**
>
> We really appreciate Reviewer ajp3 for the positive comments on our paper. We address the raised concerns below.
>
> **[W1] Details of the acquisition function.**
>
> For the acquisition function, we use Thompson Sampling, following other existing latent Bayesian optimization works [1-3]. We added detailed information about the acquisition function in Section 5.3 of the paper.
>
> **[W2] Unclear relation to Ho et al. (2019) in L255.**
>
> We refer Ho et al. (2019) since it employs a variational distribution to approximate the Evidence Lower Bound (ELBO), similar to the approach used in our NF-BO.
>
> **[W3] Explanations of embedding** $\mathbf{e}$.
>
> The latent embedding $\mathbf{e} \in \mathbb{R}^{\left\lvert \mathcal{V}\right\rvert \times F}$ consists of $\left\lvert \mathcal{V}\right\rvert$ embedding vectors $\boldsymbol{e}_j \in \mathbb{R}^F, j \in 1, 2, \dots \left\lvert \mathcal{V}\right\rvert$. Each embedding vector $\mathbf{e}_j$ corresponds to the $j$-th alphabet in the vocabulary set $\mathcal{V}$. These embedding vectors are initialized with a normal distribution. We added the further details of the embedding in our main paper.
>
> **[Q1] Application of TACS to VAE.**
>
> Good question. Our TACS (Token-level Adaptive Candidate Sampling) approach is specifically designed to operate on token-level latent vectors. However, applying TACS to a VAE is infeasible because the latent space of VAE is sequence-level, not token-level. That is why we did not include the results for VAE with TACS.
>
> [1] Eriksson, David, et al. "Scalable global optimization via local Bayesian optimization." NeurIPS 2019.
>
> [2] Maus, Natalie, et al. "Local latent space bayesian optimization over structured inputs." NeurIPS 2022.
>
> [3] Lee, Seunghun, et al. "Advancing Bayesian optimization via learning correlated latent space." NeurIPS 2014.

---

> > ### Comment · Reviewer_ajp3 · 2024-11-22
> >
> > Thank you for these clarifications.

---

### Official Review · Reviewer_2nEz · 2024-11-06

**Soundness:** 3
**Presentation:** 4
**Contribution:** 3
**Rating:** 8
**Confidence:** 5

**Summary:**

This paper addresses the value discrepancy problem in latent Bayesian optimization, where improvements to the latent variable objective value can fail to translate to improvements to objective value of the actual decision variables after decoding. The authors propose to address this issue by parameterizing the generative model as an invertible mapping, specifically a normalizing flow.

**Strengths:**

- The authors tackle an important problem in the latent BayesOpt literature, a problem which has been previously been observed since at least [1].

- The TACS procedure for selecting token positions to vary is very sensible and closely resembles a similar technique employed by MaskGIT [2] and LaMBO-2 [3], albeit with some minor differences in what signal is normalized into a sampling distribution

References

- [1] Stanton, S., Maddox, W., Gruver, N., Maffettone, P., Delaney, E., Greenside, P., & Wilson, A. G. (2022, June). Accelerating bayesian optimization for biological sequence design with denoising autoencoders. In International Conference on Machine Learning (pp. 20459-20478). PMLR.

- [2] Chang, H., Zhang, H., Jiang, L., Liu, C., & Freeman, W. T. (2022). Maskgit: Masked generative image transformer. In Proceedings of the IEEE/CVF Conference on Computer Vision and Pattern Recognition (pp. 11315-11325).

- [3] Gruver, N., Stanton, S., Frey, N., Rudner, T. G., Hotzel, I., Lafrance-Vanasse, J., ... & Wilson, A. G. (2024). Protein design with guided discrete diffusion. Advances in neural information processing systems, 36.

**Weaknesses:**

This is a reasonable paper, if somewhat incremental and with an incomplete view of previous literature on the topic. For an example of a good literature review on this area, see [4].

While more test functions and baseline experiments could always be added, I want to focus my feedback on the lack of a key experiment that verifies the authors are correctly attributing their improved performance to the elimination of value discrepancy. I find it to be a reasonable hypothesis, but the current comparisons to baselines do not prove the hypothesis to be correct because the experiment is confounded by all the other differing implementation choices in those baselines. This is not about proving that your program _as a whole_ outperforms other programs on the benchmarks, this is about critically testing your hypothesis that value discrepancy is the causal factor in this experiment. I would encourage you to think carefully about how you might design an ablation to evaluate this hypothesis. It could be as simple as making the smallest possible change to your current code to intentionally reintroduce the value discrepancy problem, while minimizing any other potential confounders in the experiment.

It is also a bit concerning that the value gap between actual and latents does not seem to have been systematically measured and reported, although that could simply be a choice of presentation. Unless I am mistaken the only place the value gap is quantified at all is in an illustrative figure in the intro.

You should also compare to the simplest possible solution to the value discrepancy problem which was employed by [1] and [3], which is simply decoding frequently and checking the actual objective value [1], and reinitializing the latent variables from actual improved solutions [3].


References

- [4] González-Duque, M., Michael, R., Bartels, S., Zainchkovskyy, Y., Hauberg, S., & Boomsma, W. (2024). A survey and benchmark of high-dimensional Bayesian optimization of discrete sequences. arXiv preprint arXiv:2406.04739.

**Questions:**

- The tradeoff of the restrictive parameterization of a normalizing flow is typically a loss of expressivity. I believe this is why normalizing flows have largely been abandoned by most of the generative modeling community. Are the authors at all concerned about this tradeoff?

- Have used techniques like TACS for some time, I've noticed it has a weakness in the exploration context, namely that it will tend not to choose positions that have not been varied in the past, and is thus heavily reliant on a relatively complete characterization of the search space in the initial data package. Have you considered how you might "cold start" your solver when only a small local region of the input space is represented in the initial dataset?

- Similarly the choice of temperature in the TACS procedure is a critical hyperparameter that was presumably tuned by the authors by looking at performance curves on the evaluation tasks. While this is common practice in the literature, it tends to cause the actual performance of these algorithms to be overestimated when turned to real consequential tasks where little or no tuning budget is available. Have the authors considered any heuristics for choosing this hyperparameter, perhaps in an adaptive way?

- If you are interested in more potential test functions and baselines you may want to check out the [poli](https://github.com/MachineLearningLifeScience/poli) and [poli-baselines](https://github.com/MachineLearningLifeScience/poli-baselines) project. In particular you may be interested in Ehrlich test functions [5], which are available [standalone](https://github.com/prescient-design/holo-bench) or as part of the poli package. Ehrlich functions are inexpensive yet relatively difficult closed-form test functions for rapid prototype development and more extensive ablation experiments.


References

- [5] Stanton, S., Alberstein, R., Frey, N., Watkins, A., & Cho, K. (2024). Closed-Form Test Functions for Biophysical Sequence Optimization Algorithms. arXiv preprint arXiv:2407.00236.

---

> ### Author Response · Authors · 2024-11-22
> **Response to Reviewer 2nEz (1/2)**
>
> Thank you for the feedback. We complement Related Works section to include the suggested survey literature [1] and incorporate additional references [2-7] as part of our revisions.
>
> **[W1] Ablation Study of SeqFlow for Value Discrepancy Problem.**
>
> As suggested, we additionally conduct an ablation study of our generative model (SeqFlow) to demonstrate the impact of the value discrepancy problem. We compare generative models by applying different mapping functions to the same normalizing flow framework: Equation (8, 9) (ours) and BiLSTM (TextFlow [8]). TextFlow [8] does not ensure the accurate reconstruction of the inputs since it applies BiLSTM to the mapping function. The optimization results are in Table 1. Please note that we do not apply TACS solely to compare generative models. From the table, our SeqFlow model achieves better performance with fewer parameters compared to the baseline model. This shows that addressing the value discrepancy problem is important in effective Bayesian optimization.
>
> **Table 1: Optimization performance on two Guacamol tasks.**
>
> | Method | **SeqFlow (Ours)** | **TextFlow [8]**  | **SELFIES VAE** |
> | --- | --- | --- | --- |
> | **BO** | **NF-BO w/o TACS** | **NF-BO w/o TACS** | **LOL-BO** |
> | **Base Model** | **Autoregressive NF** | **Autoregressive NF**  | **SELFIES VAE** |
> | **Mapping Functions** | **Eq. (8, 9)** | **BiLSTM** |  |
> | **Complete Reconstruction** | **O** | **X** | **X** |
> | adip | **0.778** $\pm$ **0.016** | 0.716 $\pm$ 0.017 | 0.716 $\pm$ 0.003 |
> | med2 | **0.372** $\pm$ **0.012** | 0.347 $\pm$ 0.010 | 0.325 $\pm$ 0.004 |
>
> **[W2] Measurements of the value discrepancy between actual and latents**
>
> To demonstrate that our SeqFlow effectively addresses the value discrepancy problem, we measure the ratio of instances where $y\ne\hat{y}$, comparing the score of the input data $y$ and the reconstructed data $\hat{y}$. We use top 1,000 data points from 10,000 initial data points across all Guacamol tasks. The experimental results are in Table 2. From the table, our SeqFlow model accurately reconstructs every data point, unlike the TextFlow model, which indicates that our SeqFlow model is appropriate NF model to address the value discrepancy problem.
>
> Moreover, we add visualizations of the value discrepancy problem on other Gucamol Tasks in the appendix to show that the value discrepancy problem generally arises.
>
> **Table 2: Quantitive measurement of value discrepancy using the ratio of** $y \ne \hat{y}$.
>
> | Model | SeqFlow | TextFlow [8] |
> | --- | --- | --- |
> | adip | 0.000 | 0.548 |
> | med2 | 0.000 | 0.609 |
> | osmb | 0.000 | 0.630 |
> | pdop | 0.000 | 0.502 |
> | rano | 0.000 | 0.814 |
> | zale | 0.000 | 0.750 |
> | valt | 0.000 | 0.001 |
>
> **[W3] Compare with methods [4] and [5] from the perspective of the value discrepancy problem.**
>
> Thanks for introducing these valuable papers. We add descriptions of papers [4,5] to the Related Works section (L109) of the revised paper and will expand the discussion in the final version. The decoder from these methods (*e.g.,* MAE) differs from our problem statement's decoder $\mathbf{x}=p_\theta(\mathbf{z})$ (Eq. 6) because they utilize the original $\mathbf{x}$ in the decoder to restore inputs for unmasked input tokens, thus cannot define the value discrepancy problem, which is defined as
>
> $$
> \mathbf{x} \neq \hat{\mathbf{x}} \text{ where } \hat{\mathbf{x}}:=p_\theta(q_\phi(\mathbf{x})), \text{and } f(\mathbf{x}) \neq f(\hat{\mathbf{x}}).
> $$
>
> Instead, these models elegantly align the latent space and input space by repeatedly decoding or encoding new candidates. One key difference between [4,5] and ours is that our method does not need additional encoding and decoding of all candidates. Our method decodes only the latent vector chosen by the acquisition function thanks to one-to-one mapping between the input and latent space.
>
> **[Q1]** **Expressivity of Normalizing Flows.**
>
> Although normalizing flows have tradeoffs between expressivity and restrictive parameterization, their injectivity property is crucial for addressing the value discrepancy problem in latent Bayesian optimization. Moreover, we find that the expressivity of NF models is sufficient to perform the generation and Bayesian optimization on *de novo* molecular optimization tasks. We think that a model with improved expressivity (*e.g.*, diffusion model) could potentially achieve superior optimization performance, provided it guarantees complete reconstruction.

---

> ### Author Response · Authors · 2024-11-22
> **Response to Reviewer 2nEz (2/2)**
>
> **[Q2] Exploration abilities of TACS according to the number of initial points.**
>
> Thank you for the thoughtful question. TACS adaptively samples the "latent" token based on point-wise mutual information between the latent token and the input sequence. This mechanism enables TACS to select the latent token that diversifies the given sequence, thereby accelerating the exploration. To verify the exploration abilities and effectiveness of our TACS, we conduct an ablation study of TACS using 1 initial data point and 10,000 initial data points in Table 3. Each experiment is repeated 5 times and we report the average and standard deviation of the results. From the table, NF-BO with TACS consistently shows better optimization results compared to NF-BO without TACS when using 1 initial data point. Moreover, NF-BO with TACS is shown to be robust to the number of initial points by comparing the optimization results between NF-BO w/ TACS (init 1) and NF-BO w/ TACS (init 10K). This suggests that TACS has strong exploration capabilities. Interestingly, in some tasks like Adip, NF-BO w/ TACS (init 1) performs better than NF-BO w/ TACS (init 10K), which highlights the effectiveness of TACS under low-data scenarios. We maintained the TACS temperature at 400, consistent with our main experiments.
>
> **Table 3: Performance on Low-Data Scenarios with 1 Initial Data.**
>
> | Method | NF-BO w/o TACS (init 1) | NF-BO w/ TACS (init 1) | NF-BO w/ TACS (init 10K, in paper) |
> | --- | --- | --- | --- |
> | adip | 0.765 + 0.038 | **0.818 + 0.051** | 0.809 + 0.059 |
> | med2 | 0.306 + 0.014 | **0.307 + 0.027** | 0.380 + 0.014 |
> | osmb | 0.848 + 0.037 | **0.855 + 0.007** | 0.897 + 0.016 |
> | pdop | 0.564 + 0.045 | **0.623 + 0.043** | 0.759 + 0.023 |
> | rano | 0.846 + 0.019 | **0.848 + 0.021** | 0.941 + 0.006 |
> | valt | 0.198 + 0.443 | **0.786 + 0.439** | 0.995 + 0.004 |
> | zale | 0.586 + 0.013 | **0.589 + 0.033** | 0.760 + 0.012 |
>
> **[Q3] Choice of TACS Temperature.**
>
> To choose the TACS temperature, we initially conduct a simple search for the TACS temperature on one of the Guacamol tasks within the range [400, 200, 100]. Based on this search, we fix the temperature at 400 for all benchmarks and tasks.
>
> To further demonstrate the robustness of the TACS temperature, we provide a sensitivity analysis in Table 4. This analysis is performed across seven Guacamol tasks, with results averaged over five runs per task and summed. Both the oracle budget and the number of initial data were set to 10,000. From the table, TACS temperatures above 200 consistently show better optimization results compared to not using TACS, highlighting the robustness of our approach to the choice of TACS temperature across all tasks.
>
> **Table 4: Sensitivity Analysis of TACS Temperature on 7 Guacamol tasks.**
>
> | TACS Temperature $\tau$ | Score sum on 7 Guacamol tasks |
> | --- | --- |
> | $\infty$ (w/o TACS) | **5.495** |
> | 2000 | **5.495 (+0.000)** |
> | 1000 | **5.523 (+0.028)** |
> | 400 | **5.544 (+0.049)** |
> | 200 | **5.565 (+0.070)** |
> | 100 | 5.481 (-0.014) |
> | 50 | 5.453 (-0.042) |
> | 20 | 5.418 (-0.077) |
>
> **[Q4] More experimental results on other test functions and baselines.**
>
> Thank you for recommending additional good optimization tasks for model evaluation. We will update the experimental results of our methods on these tasks and incorporate them into our experiments section as soon as possible.
>
> [1] Lu, Xiaoyu, et al. "Structured variationally auto-encoded optimization." ICML 2018.
>
> [2] Notin, Pascal, José Miguel Hernández-Lobato, and Yarin Gal. "Improving black-box optimization in VAE latent space using decoder uncertainty." NeurIPS 2021.
>
> [3] Verma, Ekansh, Souradip Chakraborty, and Ryan-Rhys Griffiths. "Highdimensional Bayesian optimization with invariance." ICML Workshop 2022.
>
> [4] Stanton, Samuel, et al. "Accelerating bayesian optimization for biological sequence design with denoising autoencoders." ICML 2022.
>
> [5] Gruver, Nate, et al. "Protein design with guided discrete diffusion." NeurIPS, 2024.
>
> [6] Maus, Natalie, et al. "Discovering many diverse solutions with bayesian optimization." AISTATS 2023.
>
> [7] González-Duque, Miguel, et al. "A survey and benchmark of high-dimensional Bayesian optimization of discrete sequences." arXiv 2024.
>
> [8] Ziegler, Zachary, and Alexander Rush. "Latent normalizing flows for discrete sequences." ICML 2019.

---

> > ### Comment · Reviewer_2nEz · 2024-11-22
> > **score improved**
> >
> > thanks for a thoughtful and well-executed rebuttal. I find your additional experiments very convincing and will recommend acceptance. Cheers!

---

> > > ### Author Response · Authors · 2024-12-03
> > > **Response to Reviewer 2nEz**
> > >
> > > We appreciate Reviewer 2nEz for the positive review. As previously mentioned in the response to **Q4**, we update the additional experimental results of NF-BO on suggested benchmark, Ehrlich function.
> > >
> > > **[1] Additional results on Ehrlich function.**
> > >
> > > We validate the effectiveness of our NF-BO with the Ehrlich function [1] using the poli library [2], under a 100K oracle budget, starting with a single initial data point. The parameters for the function are consistent across all runs: sequence length = 128, number of motifs = 4, motif length = 8, and quantization parameter = 8. Both the function and the initial dataset are fixed to ensure reproducibility. Our NF-BO is pretrained on 100K feasible data points. Table 5 includes the mean and standard deviation across five runs. The results demonstrate that our NF-BO consistently improves the score, while the baseline genetic algorithm (DiscreteGA, poli [2]) fails to improve the score. This indicates that our NF-BO is effective on the other benchmark tasks.
> > > We will include these experimental results in the final version of the paper.
> > >
> > > **Table 5. Performance on the Ehrlich function.**
> > >
> > > | Oracle calls | NF-BO | DiscreteGA |
> > > | --- | --- | --- |
> > > | 1 | 0.013 | 0.013 |
> > > | 10K | 0.181 + 0.049 | 0.013 + 0.000 |
> > > | 30K | 0.444 + 0.163 | 0.013 + 0.000 |
> > > | 50K | 0.530 + 0.195 | 0.013 + 0.000 |
> > > | 70K | 0.569 + 0.168 | 0.013 + 0.000 |
> > > | 100K | 0.591 + 0.188 | 0.013 + 0.000 |
> > >
> > >
> > > [1] Stanton, Samuel Don, et al. "Closed-Form Test Functions for Biophysical Sequence Optimization Algorithms." ICML 2024 W.
> > >
> > > [2] González-Duque, Miguel, et al. "poli: a libary of discrete sequence objectives." [Computer software].

---

### Author Response · Authors · 2024-11-22
**General Response**

We appreciate all the reviewers for their thoughtful comments and constructive feedback. We sincerely thank the time and effort in reviewing our paper. We provide a global overview and major updates in our paper below.

- **[Code availability]** For reproducibility, we provide both the code and the pretrained model checkpoint at the following anonymous GitHub link: https://anonymous.4open.science/r/NFBO-EA7C/
We will make the GitHub link public in the final version of the paper.
- **[Additional Experiments]** We performed additional experiments and added the results to the appendix of the main paper:
    - **More experimental results on value discrepancy problem**
    - **Ablation study and more analysis of our Token-level Adaptive Candidate Search**
- **[Revisions for Clarity]**  To improve clarity and address reviewers' comments, we revised and updated several sections of the paper, including Figure 4. Changes and additions are marked in red for easy identification.

We have carefully addressed the concerns raised during the review and hope that our responses clarify any remaining issues. If there are further concerns or questions, we are happy to address them. Detailed responses to individual comments can be found below.

---

### Meta-Review · Area_Chair_ScPo · 2024-12-19

**Metareview:**

This paper proposes to use normalizing flows which, because they are invertible, solve the mismatch problem where the data the surrogate model is trained on in latent space Bayesian optimization does not necessarily decode to the observed values being trained on. This problem leads to large inefficiencies in methods like LOL-BO because they must repeatedly re-encode data to mitigate, but not fully address, this issue. The authors' method clearly enjoys overwhelming data efficiency advantages, and in some cases even y axis improvements at "full" data budgets.

This is a great paper and an obvious accept, both to all of the reviewers and to me.

**Additional Comments On Reviewer Discussion:**

All reviewers were convinced by extensive additional results. Looking through the updated draft of the paper, it has clearly been updated extensively with changes highlighted in red.

---

### Decision · Program_Chairs · 2025-01-22

Accept (Oral)